# Aerogels Part 2. A Focus on the Less Patented and Marketed Airy Inorganic Networks Despite the Plethora of Possible Advanced Applications

**DOI:** 10.3390/ijms262110696

**Published:** 2025-11-03

**Authors:** Silvana Alfei

**Affiliations:** Department of Pharmacy (DIFAR), University of Genoa, Viale Cembrano, 4, 16148 Genoa, Italy; alfei@difar.unige.it; Tel.: +39-010-355-2296

**Keywords:** aerogels (AGs), inorganic-based AGs, chalcogenide AGs, metal AGs, noble metal AGs, sol–gel synthesis, not sol–gel routes, templating methods

## Abstract

Recently, the state of the art of aerogels (AGs) has been reviewed, reporting first on their classification, based on the chemical origin of their precursors and the different methods existing to prepare them. Additionally, AGs of inorganic origin (IAGs) were contemplated, deeply discussing the properties, specific synthesis, and possible uses of silica and metal oxide-based AGs, since they are the most experimented and patented AGs already commercialized in several sectors. In this second part review, IAGs are examined again, but chalcogenide and metals AGs (CAGs and MAGs) are debated, since they are still too little studied, patented, and marketed, despite their nonpareil properties and vast range of possible applications. First, to give readers unaware of the previous work on AGs, a background about IAGs, all their main subclasses have been reported and their synthesis, including sol–gel, epoxide addition (EA), and dispersed inorganic (DIS) methods, as well as procedures involving the use of pre-synthesized nanoparticles as building blocks, have been discussed. Morphology and microstructure images of materials prepared by such synthetic method have been supplied. Conversely, the methods needed to prepare CAGs and MAGs, topics of this study, have been debated separately in the related sections, with illustrative SEM images. Their possible uses, properties, and some comparisons of their performance with that of other AGs and not AG materials traditionally tested for the same scopes, have also been disserted, reporting several case studies in reader-friendly tables.

## 1. Introduction

Aerogel is an open, non-fluid colloidal network or polymer network that is expanded throughout its whole volume by a gas and is formed by the removal of all swelling agents from a gel without substantial volume reduction or network compaction. Particularly, liquids are replaced with air using different drying techniques, whose method strongly contributes to the physical characteristics of resulting materials [1]. Generally, the best materials depend on the synthetic method employed and possess an exceptionally low density (0.0011–0.5 g/cm^3^), great specific surface area (SSA), and can be extremely easily spread through their 3D network. Additionally, in these materials, thermal conductivity, acoustic velocity, refractive index, and dielectric constant are very low [2]. In NASA planetary rowers, subsea systems, industrial conduits, oil refineries, constructions, fridges, and dresses, AGs have primarily been used as thermal insulators [3]. However, AGs have been shown to be suitable for several other applications [2,3]. Collectively, they may be promising catalysts, catalyst supports [4,5,6], sensors, interlayer dielectrics and optical applications [7], filters [8], collectors for cosmic dust [9,10], detectors in particle physics [11,12], thermal insulators [13], etc. [7,14,15,16,17]. Moreover, AGs could be used to engineer batteries, capacitors, and as constituents in fuel or solar cells [18]. Unfortunately, notwithstanding these possibilities, the expensive precursors necessary to synthesize them, the difficulty to control the sol–gel procedures mainly utilized for their preparation [19], and the need of particular and more complicate synthetic methods for preparing some classes of them, such as chalcogenide AGs (CAGs), as well as pure noble metal and metal AGs (NMAGs and MAGs), strongly limit the translation of the most challenging ones from laboratory setting into the market. In fact, while silica AGs (SAGs) and metal oxide AGs (MOAGs) have been on the market for years, especially for thermal and acoustic insulation, CAGs, NMAGs, and MAGs remain mainly limited to laboratory experimentation. Among other reasons, there are the more complex, expensive, and poorly sustainable synthetic methods [20]. NMAGs and MAGs are regularly not directly synthesized via traditional molecular routes and need more laborious two-step and one-step sol–gel processes and hard conditions, involving thermal treatment in reducing atmospheres [20], which creates further difficulty in reaching the desired crystallinity. As for other AGs, the use of preformed nanoparticles (NPs) as building blocks to be assembled for achieving the desired network can be of help to limit this issue. Additionally, this approach can be associated with the use of templates to control the assembly behavior of NPs into the 3D percolating network, to obtain scaffolds stable enough to allow further processing [21]. This strategy represents an elegant and powerful method to obtain crystalline monolithic AGs [21]. Moreover, other non-sol–gel processes have been experimented with to prepare NMAGs and MAGs, such as dealloying, combustion, bio templating, and salt templating, which are of help but require hard work and are expensive. An exciting method to obtain MAGs consists of carbothermal reduction of oxide AGs, providing iron AGs which can function as either explosives, pyrotechnics, or thermites depending on their nanostructure. Similarly, the preparation of CAGs, intended as metal CAGs (MCAGs), requires costly precursors poorly marketed and peculiar synthetic methods, such as the thiolytic sol–gel procedure, the controlled assembly of NPs, and metathesis methods. The aid of solid templates and soft or hard templating methods is then necessary to achieve mesoporous functional CAGs (MMCAGs), which are the most performant in environmental remediation, as well as for catalysis, electrocatalysis, and energy generation. Nowadays, based on the chemical origin of their precursors, three great families of AGs are known, including inorganic, organic, and composite AGs [22]. Figure 1 offers a schematic representation of them, with related subclasses.

While in the recent part one of this work, IAGs have been reviewed, with particular attention to SAGs and MOAGs, which for years have been translated from the laboratory experimentation to the market, it was important to create the second part of this study on IAGs. This allows for further discussion on the current state of the art of CAGs and MAGs, whose patents are limited and marketing trivial, despite their nonpareil properties and huge possible applications. To this end, after introducing new readers who are unaware of the first part of the paper, a brief background on IAGs, summarizing their main conventional synthetic methods, as well as supplying images of the morphology and microstructure achievable using these approaches, MCAGs, MMCAGs, NMAGs, and MAGs are discussed. The peculiar methods needed for the preparation of the above-mentioned structures are debated in detail in the related sections, supplying illustrative SEM images. Their possible main uses, properties, and comparisons of their performance with that of other materials traditionally used for the same scopes were also discussed, reporting several case studies, schematized in reader-friendly tables. The main scope of this second-part paper is to provide more visibility to CAGs, NMAGs, and MAGs, to stimulate more experimentation on them, both in terms of synthesis and applications, with a final outcome of patent development. Augmenting such type of research on them, too limited so far, their easier and less expensive production, as well as their more rapid translation on the market, could occur.

## 2. Subclassification of Inorganic AGs

As summarized in Table 1 below, inorganic AGs, which are one of the three classes of AGs based on the chemical origin of their precursors, comprise several subclasses.

Collectively, IAGs include SAGs, CAGs, OAGs, CBAGs, NAGs, MAGs, and NMAGs [2]. These AGs can be synthesized, starting from metal alkoxides or metal salt compounds as inorganic precursors, following either traditional or specific synthetic methods.

CBAGs comprehend SC/AGs, CDC/AGs, and MC/AGs. CBAGs have been widely examined in the area of catalytic production of hydrogen, as electrodes in supercapacitors, as components in gas turbines, as well as for heat dissipation and exchangers [29,30,31]. In addition, CHC/AGs, HAN/AGs, TIC/AGs, and TUC/AGs have been reported. NAGs include mainly CN/AGs, ALN/AGs [32,33], BN/AGs, and SIN/AGs. Among them, BN/AGs are typically used in photocatalysis, engineering phase change materials, and environmental remediation by pollutant removal or degradation [34,35].

While SAGs and OAGs have been widely discussed in the first part review born from my project about IAGs recently reported [37], chalcogenide-based AGs (CAGs) or properly metal chalcogenide AGs (MCAGs), as well as MAGs and NMAGs, are the topic of this second part review. CAGs and MCAGs are made from chalcogenides [24], which are compounds encompassing a chalcogen anion and a more electropositive element. MCAGs are preferentially applied to absorb heavy metals from water [38], such as mercury, lead, and cadmium [38]. Specifically, MCAGs of non-platinum metals are very good for desulfurization [39].

Furthermore, mesoporous MCAGs (MMCAGs), which are MCAGs endowed with high SSA, are most encouraging for gas separation [40,41] and for the removal of radionuclides such as ^99^Tc, ^238^U, and ^129^I, from nuclear waste [42].

Pure metal AGs (MAGs) and noble metal AGs (NMAGs) represent a singular class of inorganic AGs owing to the properties of both metals and ultralight and low-density materials, widely applied in detection-based sectors [36,43]. MAGs and NMAGs are mainly prepared as networks of metal NPs in colloidal suspensions, which are dried using supercritical CO_2_ to produce nanofoams mainly of iron, gold, and silver [36].

## 3. Conventional Synthetic Methods to Achieve AGs

In this Section, the conventional synthetic methods to achieve AGs, different from MCAGs, MMCAGs, and MAGs, have been reported and discussed. Real images of the morphology and microstructure of AGs obtained by such methods have been provided in Appendix A, which are included in the Appendix A. Numerical citations of papers from which Appendix A and related discussions present in this section have been taken are reported in their captions, respecting the numbers reported in the Appendix A description at the end of this paper, which follows the order of the quotations present in the main text. Specifications about the copyright license needed have also been included in the Figures captions. Readers can find references of such citations in the reference list after the last reference quoted in the main text. AGs are commonly prepared starting from different molecular precursors, following the aqueous sol–gel method [37,44]. It encompasses different stages, including first the hydrolysis of precursors in acidic or basic conditions, followed by the polycondensation of the products [45] via crosslinking reactions [37]. This step leads to the obtainment of an open-porous network, containing a continuous liquid phase known as hydrogel (Figure 2) [37,46].

Once the gel network is formed, to obtain a mechanically more stable scaffold, it is often aged and ripened before drying [47]. In this regard, the type of solvent used in the gel network formation step can significantly affect the aging process and physicochemical properties of aged samples and AGs. When gels were formed and aged in tetraethyl orthosilicate (TEOS), water, or methanol, at different temperatures and for different times, AGs with densities in the range of 0.1–0.6 g/cm^3^ were achieved. The compression properties of the gel increased with increasing aging time and temperature, while solvents with higher polarity improved polymerization, which enhanced the mechanical properties of the related samples. However, when water was used, the stresses and capillary forces during atmospheric drying were very large, thus inhibiting the “spring-back effect” and consequently a collapsed silica network with higher density was fabricated. For methanol, alcohol inhibits the reactions inconveniently, causing more shrinkage. In aging by n-hexane, capillary pressure declined significantly, and thereby, shrinkage was eliminated, and AGs with low bulk densities (0.095 g/cm^3^), high specific surface areas (600 m^2^/g), and large pore volumes (2.6 cm^3^/g) were synthesized. Then, the drying procedure may occur by ambient pressure drying, supercritical drying, or freeze drying. Collectively, drying methods constitute an essential step in AGs fabrication, directly determining the pore structure, thus affecting their properties and functionality. Appendix A shows how the various drying methods can determine pore structure modifications in AGs, thus regulating their microstructure, density, and performance characteristics. When the freeze-drying method is used, the volume expansion of water during freezing and the compression of structure by ice crystals modify the original pore size and morphology, but a macro-structured architecture with large macro pores is reconstructed after the process (Appendix A). In the supercritical CO_2_ drying, water present in the hydrogel is replaced by anhydrous ethanol (EtOH) before drying. EtOH extraction by supercritical CO_2_ enables the avoidance of the pore collapse phenomenon in order to keep the porous texture of the wet material intact, and only trivial changes can be observed (Appendix A). For samples derived from ambient vacuum drying (VD-A) (Appendix A), oven drying (OD-A) (Appendix A), and natural drying (ND-A) (Appendix A), water is exchanged with EtOH, like in the previous case. Due to the different drying conditions, EtOH generates surface tension during drying, thus triggering capillary forces that transform the pristine wood pore structure. Specifically, in ND-A, the liquid surface tension provokes the most remarkable structural transformations and causes the most important volume shrinkage (Appendix A). Collectively, supercritical CO_2_ drying excellently maintains the original microstructure, providing AGs with a major thermal insulation capacity. In the freeze-drying process, ice crystals form, from which the original macrostructure can be rebuilt, providing AGs with high porosity, very low density, notably improved permeability, and optimal cyclic water absorption capacity (WAC). Collectively, both VD-A, OD-A, and ND-A cause important deformation of the original pore structure. OD-A increases the original number of pores via volumetric contraction, thus providing the highest SSA. On the contrary, ND-A provides AGs with the highest density and the worst thermal insulation capacity. On these considerations, it appears that supercritical CO_2_ drying represents the most critical stage of AGs preparation. This drying method allows for maintaining the three-dimensional (3D) pore structure of hydrogels, thus leading to nonpareil material properties, including high porosity, large SSA, and very low density [37,48]. After aging, the supercritical CO_2_ drying of hydrogels can be carried out both at high and low temperatures and in an autoclave under reduced pressure [37]. During this process, surface tension in the gel pores and constant compression, characterizing other drying procedures, do not occur, thus preventing the gel body from collapsing due to the forces at the three-phase interface [37,45]. The drying rate of AGs strongly depends on the drying method. Specifically, FD-A exhibited the longest drying duration (22 h) and the lowest drying rate. ND-A required 5 h for complete drying, which was longer than that of OD-A. Conversely, VD-A exhibited superior drying efficiency (3 h) compared to OD-A (3.5 h) despite identical temperature conditions. Additionally, density, which significantly influences sound absorption, thermal conductivity, mechanical properties, and water absorption performance of AGs, strongly depends on the drying method. The ascending order of WAGs density was FD-A < SCD-A < VD-A < OD-A < ND-A, indicating that the drying method directly determines the final density. Furthermore, other problems exist, which are still not completely solved, and are associated with the sol–gel synthesis of AGs. It remains very difficult for operators to control the hydrolysis and condensation rates, which are too rapid, thus making it difficult to adjust the gelling behavior and tune the final porosity of AGs [37]. The advent of the epoxide addition (EA) method has promoted a notable advancement in the sol–gel process in the preparation of AGs. The EA method has allowed for slowing down the hydrolysis and condensation rates, thus obtaining material with high porosity [47]. The EA method using polyethylene oxide (PO) represents a modification of the traditional sol–gel alkoxide method reproposed by Gash et al. to prepare Fe_2_O_3_ [48] and Cr_2_O_3_ [49] AGs [37]. Inorganic salts were used as precursors, while acidic hydrolysis and ethylene oxide derivatives were performed and adopted to obtain colloidal dispersions and induce gelation. By lowering the kinetics of the hydrolysis and condensation steps, the formation of precipitates commonly obtained by the traditional sol–gel method was avoided, thus achieving monolithic AGs [37]. Figure 3, reproducing that reported in our first-part paper on IAGs [37], shows the various types of epoxides which have been used to prepare iron oxides AGs [37,50].

Figure 4 shows the recognized mechanism of the most reported EA approach, in this case involving the use of propylene oxide (PO) for gel formation in acidic conditions [37].

Figure 4A schematizes the first step, where the oxygen of PO is protonated by the H^+^ of HA acidic catalyst necessary for the sol–gel process, while Figure 4B shows the action of anion A^−^, making a basic attack on the less substituted carbon atom of PO, leading to a ring opening reaction [37]. This approach captures acidic proton atoms, thus slowly the rais of the pH value. Such a pH change leads to hydrolysis and condensation of the inorganic salts and slows down the condensation rate [37]. The studies of Gash et al. inspired other researchers to use the EA method, thus achieving many different unprecedented materials, significantly escalating the range of possible monolithic AG materials achievable. Specifically, the pioneer studies of Gash et al. on Fe_2_O_3_ were then progressively enlarged to Fe_3_O_4_, β-FeOOH [49], Fe_3_C [51], Pd- and K-doped iron oxide and iron–chromium oxides used in a mixture [50]. Before 2020, several materials were used for the preparation of AGs via the EA method, which are reported in Table 2.

A further enhancement in the sol–gel technique, which allowed to produce crack-free monolithic materials from bivalent metal ions (Fe^2+^, Cu^2+^, Ni^2+^, and Zn^2+^), was achieved by the discovery of the dispersed inorganic method (DIS). Du et al. proposed that DIS is a further evolution of the EA procedure, in which polyacrylic acid (PAA) and propylene oxide (PO) are added to inorganic salt solutions [72]. The newly added component (PAA) acts both as a dispersant, due to its steric hindrance, and as a nucleation site for gel formation [37]. The DID method allowed for the preparation of AGs containing bivalent metal ions, by easy 3D crosslinking reactions, characterized by less shrinkage and stronger scaffolds [37,72,73]. However, AGs can also be manufactured using several types of currently available NPs having different sizes and shapes due to decades of research in nanotechnology, as versatile building blocks with unique properties and functionalities [37]. NPs have been assembled as LEGO bricks in top-down processes [37,74] to form monolithic AGs, in which the structure of the NP properties was conserved in a nanostructured macroscopic bulk material [37]. Unfortunately, AGs prepared using NPs show lower porosity and SSA, while higher-density AGs are differently prepared. However, such compounds possess unprecedented properties for AGs, including super-paramagnetic effects [75], ferroelectricity [76], luminescence [77], (photo)catalytic activity [78,79], or electrical conductivity [80]. The new functionalities of NP-based AGs, derived from the specific type of NPs used for their synthesis, make them suitable for engineering batteries [81], fuel [82], or solar cells [83]. Briefly, the preformed nanocrystal building blocks are dispersed in proper solvents in high concentration. Then, the gelation of NP dispersion, to finally form 3D macroscopic AG monoliths, commonly occurs by controlled destabilization. This stage causes the assembly of NPs in hydrogels at random crystallographic orientation, as in the case of MCAGs, or by an oriented attachment mechanism, as in the case of TiO_2_ and SnO_2_ AGs [37]. Moreover, NP building blocks dispersion can be destabilized via mild centrifugation, as in the case of WO_x_ and Y_2_O_3_ AGs [37]. Moreover, nanosized building blocks have been previously prepared by wet chemical processes, by controlling their particle size, size distribution, shape, and surface chemistry, thus preventing or minimizing agglomerate formation and ultimately making it easy for re-dispersion in the next step [37]. The available wet synthetic methods encompass aqueous [84] and nonaqueous sol–gel processes [83,85], polyol route [86], hot-injection [87], heating-up method [88], hydro- and solvothermal processing [89], etc. [37]. Once synthesized, the NP building blocks are dispersed in solvents at a concentration sufficient to reach a percolation threshold during gelling [37]. Excessive concentration should be avoided to prevent undesired aggregation, which can be averted by using brushes, surfactants, or electronic charges [37]. Generally applicable protocols to prepare colloidal NPs dispersions at an ideal concentration are still missing, and research in this sense is intense [37]. The ideal combinations of stabilizing approaches and solvents, and the correct balance of attractive and repulsive forces between particles, are peculiar for a specific system and determine if it will evolve to give a stable dispersion or undergo coagulation [37,90,91,92]. Concerning the following destabilization step of the dispersions, the most adopted strategies consist of photochemical treatments [93], temperature changes [81], sonication [76], or the addition of chemicals [77]. Moreover, further solvents can be added [37,80] to remove stabilizing ligands from the surface of the NPs [79] or to neutralize surface charges by changing the ionic strength [94] or pH [95] of the media [37]. It is noteworthy that the impairment of the balance between attractive and repulsive forces leads to attractive interactions to prevail [37,96]. Upon this phenomenon, NPs lose their dispersed hard sphere-like morphology, becoming “stickier”, thus colliding/fusing together [97]. At this stage, it is mandatory to control the rate of the destabilization process [98,99], to avoid flocculation and sedimentation in place of AGs formation via a percolating network formation throughout the entire volume of the sample [37,96]. The last stage of NP-based AGs synthesis consists of the super critically drying of the wet gel, which leads to AGs created by assembling building blocks a few nanometres in macroscopic structures centimeters in size [37]. Figure 5 summarizes the main conventional synthetic methods to prepare AGs.

More complex synthetic methods are needed to prepare chalcogenide AGs (MCAGs and MMCAGs) and pure metal AGs (NMAGs and MAGs), topics of this study, including the thiolytic sol–gel method, metathesis, controlled aggregation of nanoparticles (NPs) to achieve MCAGs, and soft and hard template methods to achieve MMCAGs. Two-step and one-step sol–gel processes, in hard conditions involving thermal treatments in reducing atmosphere, followed by supercritical CO_2_ drying, and not sol–gel procedures including dealloying, combustion, bio-templating, and salt-templating approaches, are employed to achieve NMAGs and MAGs. These synthetic methods will be described in detail in the section focused on these types of AGs.

### 3.1. Microstructure of AGs Prepared via the Synthetic Methods Previously Described

#### 3.1.1. Sol–Gel Methods Starting from Molecular Precursors

Appendix A demonstrates the SEM micrographs of agarose AGs (AA-2), silica AGs (SA-4), and composite AGs (CAs), prepared by the in-situ sol–gel method. The 3D network structure of AA-2 shows dispersed and disordered agarose nanofibers connected by hydrogen bonding or electrostatic attractions (Appendix A). On the contrary, SiO_2_ AG structured aggregates of micrometers in size are observable in the pure SA micrograph (Appendix A). Conversely, during the gelation process of CAs, a gel skeleton made of agarose was formed first because of its low self-coagulation temperature. Only in a second moment, SiO_2_ gelled in the agarose gel structure, thus creating an interpenetrating network (IPN) structure (Appendix A). SEM micrographs demonstrate that the IPN structure was formed by a flexible agarose and a rigid SiO_2_ gel architecture (Appendix A). Collectively, the CAs structure appears as a merging of the AA-2 and SA-4 structures in a unique architecture. Moreover, Appendix A shows (a) TEM micrographs, (b) HRTEM micrographs, (c) STEM-EDX images, and (d–i) mapping images of Ag-modified Cr-doped BaTiO_3_ (5% Ag/BTO-Cr010) AGs, prepared using a sol–co-gelation technique that involved two metallic alkoxides and a supercritical CO_2_ drying method, followed by Ag nanoparticles (Ag NPs) deposition. Particularly, Appendix A displays monodispersed BaTiO_3_ particles with a size of about 10 nm. On the contrary, the post-synthesis deposited Ag NPs demonstrated high poly-dispersion and particle size ranging from 10 to 70 nm. Appendix A exhibits the lattice appearance of a particle with a D-spacing of 0.284 nm and 0.236 nm, consistent with the cubic BaTiO_3_ structure (110 plane) and elemental silver (111 plane), respectively. Appendix A displays the STEM-EDX image of the AG, where the copper peaks fit the target stand. Ba (14.15%), Cr (1.43%), Ti (16.17%), O (59.42%), and Ag (8.84%) were the atomic percentages of all elements. Finally, the mapping images of the AGs are observable in Appendix A. The SSA of BTO-Cr0_x_ samples without deposited Ag measured using the BET method (BTO-Cr001, BTO-Cr005, and BTO-Cr010) were calculated to be 109.8, 108.8, and 107.2 m^2^/g, respectively. These high SSA values provided a larger number of surface-active sites, facilitating the migration of charge carriers and thereby enhancing photocatalytic performance. Additionally, their pore-size distribution, calculated using the classical Barrett–Joyner–Halenda (BJH) model, was estimated to be 29.8, 20.4, and 21.0 nm, respectively.

#### 3.1.2. Improved Sol–Gel Methods

##### Epoxide Addition (EA) Method

Appendix A shows the FE-SEM images of Zr–Mg mixed oxide AGs surface prepared using the EA method. Particularly, the images refer to mixed oxide AGs with 10:0 (Appendix A), 9:1 (Appendix A), 8:2 (Appendix A), 7:3 (Appendix A), and 6:4 (Appendix A) molar ratios of Zr-to-Mg, respectively. Large clusters are observed for the Zr–Mg mixed oxide AGs with a 6:4 molar ratio of Zr to Mg, thus indicating that particle aggregation increased with increasing Mg molar proportion. The SSAs and pore size distribution of the prepared Zr–Mg mixed oxide AGs with different Zr/Mg molar ratios were measured using nitrogen adsorption/desorption isotherms. The pore size distributions of the Zr–Mg mixed oxide AGs were narrow and concentrated between 2 and 10 nm, while the SSAs were 465 (Zr/Mg of 10:0), 283 (Zr/Mg of 9:1), 365 (Zr/Mg of 8:2), 371 (Zr/Mg of 7:3), and 261 (Zr/Mg of 6:4) m^2^/g, respectively.

##### Dispersed Inorganic Method (DIS)

Appendix A show the 3D microstructures and morphologies of the synthesized macro-porous Zn hydroxide monolith samples observed with SEM, prepared by using the DIS method. It consisted of a sol–gel process associated with phase separation in the presence of polyacrylic acid (PAA) and propylene oxide (PO). Appendix A show the effect of PPA, PO, solvent, and precursor amount on the macrostructure of the resulting AGs. Specifically, SEM images of AGs with varied PAA amounts ((a) 0 g, (b) 0.8 g, (c) 1.6 g, (d) 2.4 g, (e) 3.2 g, and (f) 4.0 g) are shown in Appendix A; images of (a) sol–gel transform in 30 min (up) and appearance of typical AGs (down) and SEM images of AGs with different PO contents ((b) 1.8 mL, (c) 2.2 mL, and (d) 2.6 mL) are reported in Appendix A. SEM images of AGs with varied amounts of solvents (W = water, G = glycerol) ((a) W:G = 2:1.6, (b) W:G = 1.6:2.0, (c) W:G = 1.2:2.4, (d) W:G = 0.8:2.8, (e) W:G = 0.4:3.2 and (f) W:G = 0:3.6), are pitched in Appendix A, while Appendix A shows (a) the appearance of a typical AG sample and SEM images of AGs with varied amounts of precursors added ((b) 1.22 g, (c) 1.34 g, (d) 1.46 g, (e) 1.58 g, and (f) 1.70 g). Collectively, it was observed that in the ZnCl_2_-PAA-PO system, PAA functioned as an inducer of phase separation and promoted the framework formation, thus controlling both the phase separation and the formation of macrostructures. Conversely, PO acted as a proton scavenger to trigger the gelation of the system and freeze the macrostructures. It was observed that appropriate amounts of precursor (ZnCl_2_), PAA, PO, and solvents permitted the achievement of zinc (Zn) hydroxide monoliths with continuous skeletons and interconnected macro-pores of 1 µm size.

#### 3.1.3. Nanoparticle-Based Methods

Appendix A shows the SEM image of hybrid AGs combining collagen (C) and chitosan (CH), prepared without nanoparticles (NPs) as a reference AG (Appendix A, Ref-AG) and using chemical (Appendix A, Ch-AG) and green (Appendix A, Gr-AG) iron oxide NP dispersions, previously synthesized as LEGO bricks. NPs dispersion in acetic acid was inserted in a test tube, which was in turn placed in an ultrasound machine for 15 min, to facilitate NPs dispersion. Then, it was added to the biopolymer’s solutions, after their centrifugation, and mixed before the freezing step. The three as-prepared AGs showed a rambling architecture encompassing micro- and macro-pores. The reference (Ref) AG and AGs made using chemical NPs (Ch AGs) showed similar laminar structures, with a regular directionality (Appendix A). The AGs prepared using green NPs (Gr AGs) exhibited a laminar structure, as well, but were characterized by greater heterogeneity and without directionality (Appendix A). Moreover, Ref and Ch AGs presented a multitude of pores, heterogeneous in sizes and shapes (mean porosity of 46.5 ± 15.8 nm and 29.4 ± 17.4 nm, respectively) and a relatively uniform dispersion. On the contrary, Gr AGs were almost deprived of pores, with the existing ones being uniformly minute.

## 4. More in Deep in the Less Patented Classes of Inorganic Aerogels

In this section, we extensively discuss the main equipment and possible applications of CAGs (MCAGs and MMCAGs), NMAGs, and MAGs to give them more visibility, since, according to data reported in the CAS Content Collection (Figure 2 in my recent paper [37]), they remain understudied and mainly journal articles have been published. Patents are rare and this evidence that their translation on the market is still far away, despite their nonpareil characteristics demonstrated in the laboratory experimentation. Surely, this paucity of research on these AGs compared to other and the limited scaling up of their production to have available materials in quantity sufficient for industrial production, commercialization and daily use, is due to the more complex, long, laborious, and expensive synthetic methods requiring heat treatments under reducing conditions, the use of hazardous reagents, and the poor commercialization of needed precursors to prepare them. Furthermore, only expansive research on these AGs can pave the way for novel, less expensive, safer, and more sustainable production methods. Hence, the main scope of this paper was to trigger more interest and promote this research area.

### 4.1. Chalcogenides-Based Aerogels (CAGs)

Chalcogenides-based aerogels (CAGs) are metal chalcogenide aerogels made from chalcogenides, which are chemical compounds consisting of at least one chalcogen anion, such as S^2−^, Se^2−^, Po^2−^, or Te^2−^ and at least one or more electropositive elements, such as Ti^4+^, Nb^4+^, Y^x+^, La^4+^, Mo^4+^, W^x+^, Zn^2+^, and Ge^x+^ [100]. Chalcogenide anions (S^2−^, Se^2−^, Po^2−^, or Te^2−^) possess singular properties such as direct bandgap semi-conductivity that spans the solar spectrum, accessible redox states that drive catalysis, and soft Lewis’s basicity, which instigated researcher to synthesize aerogels based on chalcogenide frameworks. Not reported until 2004, CAGs, in the past 15 years, demonstrated compositional range and properties suggesting that they could be a fertile area for study [101]. Metal chalcogenide properties combined with the high SSA and porosity, which are typical of the AGs framework make CAGs appealing for several applications. CAGs may be of great help in photoactivated processes, where they can be applied to construct solar cells, photocatalysts, sensors, etc.), in the catalytic water splitting, environmental removal of heavy metal ions, in gas separation processes, etc. Dichalcogenides encompassing transition metals, such as molybdenum (MoS_2_) and tungsten disulfides (WS_2_) possess high efficiency as electrocatalysts and photocatalysts, with regulable bandgaps and large catalytically active SSAa [102]. While carbon-based AGs typically exhibit not chemically active surfaces, thus needing cost- and time-effective extra functionalization reactions, to enhance their chemical performance, transition metal dichalcogenides are di per se catalytically active, without the need for post-synthesis functionalization [37,103,104]. Thio-lysis reactions of molecular metal precursors, condensation reactions between small negatively charged metal chalcogenides and linking cations (metathesis) or oxidative processes to form polysulfides, and condensation reactions of metal chalcogenide NPs, are the synthetic routes most used to prepare CAGs [101]. The synthetic methods can involve many sulfur-containing anions, such as tetra-thiomolybdate, and different metal ions have been used as linkers, including Co^2+^, Ni^2+^, Pb^2+^, Cd^2+^, Bi^3+^, and Cr^3+^ [24,38,39,40]. CAGs have been extensively employed as optical sensors, for CO_2_ electroreduction and for environmental remediation [105,106,107,108]. CAGs can be used for gas separation exhibiting high selectivity in CO_2_ and C_2_H_6_ rather than H_2_ and CH_4_ adsorption [39,40], thus being of great help in exiting gas stream composition of water, gas shift reaction and steam reforming reactions, widely used for H_2_ production [109]. Additionally, the realization of CO_2_-containing gas pairs separation, such as CO_2_/H_2_, CO_2_/CH_4_, and CO_2_/N_2_ is an important phase in the precombustion capture of CO_2_, natural gas sweetening and post combustion capture of CO_2_ processes, thus decreasing the amount of greenhouse gases released and mitigate the effects of climate change [110]. The above-mentioned conditioning makes the gas suitable for several applications in fuel cells. Finally, CAGs have demonstrated to be very effective in catching radionuclides from nuclear waste such as ^99^Tc, ^238^U, and especially ^129^I [41]. Although CAGs make part of a larger family of porous materials, namely porous metal chalcogenides, whose pores encompass micropores (<2 nm), mesopores (2–50 nm), and macropores (>50 nm), there is a substantial difference between mesoporous metal chalcogenide AGs (MMCAGs) and metal chalcogenide AGs (MCAGs) [107]. Table 3 summarizes the most important information on both MMCAGs and CAGs, evidencing the existing differences in terms of physicochemical properties, synthesis, structures, and applications [107].

#### 4.1.1. Synthetic Methods and Possible Applications of Metal Chalcogenide Aerogels (MCAGs) and Mesoporous MCAGs (MMCAGs)

In this Section, the synthetic methods needed to achieve MCAGs and MMCAGs have been reported and discussed. As examples, real images of the morphology and microstructure of MMAGs obtained by such methods are provided in Appendix A, included in the Appendix A file. Numerical citations of papers from which Appendix A and related discussions are presented in this Section have been taken have been reported in their captions, respecting the numbers reported in the Appendix A description at the end of the paper, which follows the order of the quotations in the main text. Specifications about the copyright license needed have also been included in the Figure’s captions. As mentioned above, readers can find the references of such citations in the reference list after the last reference quoted in the main text. Figure 6 schematically summarizes the methods needed to prepare CAGs.

Amorphous or poor crystalline metal chalcogenide sulfide hydrogels can be prepared by sol–gel methods through an initial thiolytic process of molecular metal precursors, where H_2_O is replaced by H_2_S gas, thus achieving sulfur-linked gels. Like the hydrolysis processes to achieve other hydrogels, the relative reaction kinetics of thiolytic processes and condensation play a key role in gel formation [11]. Additionally, the controlled aggregation of nanoparticles and metathesis of chalcogenidometalate are other methods used to prepare MCAGs. Metathesis, a partner-switching polymerization reaction, is a reaction towards sulfide MCAGs, where soluble chalcogenide clusters are linked by metal ions, leading to a 3D network [24,37]. Stanić et al. prepared CAGs using ZnS [122], WS_x_ [123] and GeS_2_ precursors [124,125]. An improved thiolytic process rate, leading to a grainier gel, produced doped-GeS_2_ with Er^3+^ [126]. Kanatzidis et al. succeeded in synthesizing various sulfide CAGs, observing interesting ion-exchange properties [127,128,129] and high efficiencies in the adsorption of heavy metals and gases [40,128,129,130], especially for CoMo_x_S_x_ [39,131]-based CAGs [38,132,133]. Precursor clusters were reacted in a controlled metathesis process, leading to a bottom-up assembly of redox-active species. The resulting link in a network made of tin sulfide clusters created hybrid systems, which are combined with biomimetic and porous properties of heterogeneous catalysts. Banerjee et al. reported that CAGs prepared using FeMoS inorganic clusters, efficiently photochemically reduced N_2_ to NH_3_ under white light irradiation, in aqueous media, at ambient pressure and room temperature, with promising implications in solar energy utilization [134]. CAGs which were composite of [Mo_2_Fe_6_S_8_(SPh)_3_]^3+^ and [Sn_2_S_6_]^4–^ clusters in solution, demonstrated strong optical absorption, high SSA, and good aqueous stability [135]. CAGs similarly produced demonstrated high potential in the production of solar fuels [135,136,137,138,139], while those engineered by Riley et al. were successful in the remediation of radionuclides, thus paving the way for solving problems in nuclear waste treatments [41,140,141,142]. Recently, Rothenberger et al. reported on polysulfides such as KFe_x_M_x_S_x_, where M could be Sb, As, Co, Y, or Eu [143,144].

CAGs as CuSb_2_S_4_ [145], and the first telluride-based quaternary aerogel (KFeSbTe_3_) [118], showed great potential in gas adsorption for the purification of gases. Readers particularly interested in CAGs can find additional information in several relevant reviews in this area [24,25,42,146,147,148]. In the context of MMCAGs, which have emerged as promising materials for environmental remediation and generation of sustainable energy generation, due to their adjustable optical band gap, highly polarizable surface, chemical activity, and tuneable architecture, template-assisted approaches remain the most robust methods to achieve the desired mesostructured functional materials [106]. Template-assisted synthesis consists of using a “template”, which is a critical tool that acts as a scaffold to guide the growth of mesoporous nanostructures with various geometries and morphologies [111]. Size, morphology, and charge distribution of selected template notably affect their structure-guiding capacities and the properties of the deriving MMCAGs. Template-assisted strategies include soft-templating and hard-templating (or nano casting) methods [111].

In the first approach, supramolecular aggregates, like amphiphilic surfactants or high molecular weight (HMW) block copolymers, characterized by the capability to co-assemble with organic or inorganic guest compounds, such as MCAGs nanocrystals, are used to function as templates [106,112]. Once templates are dispersed in a polar medium (water, ethanol), they self-assemble into micellar structures, which can co-assemble with metal chalcogenide nanocrystals and form liquid crystalline mesophases, thus leading to the synthesis of ordered mesoporous materials. For this behavior, such templates merit the name of structure-directing agents (SDAs) [106]. Then calcination, pyrolysis, ion-exchange, or solvent extraction are performed to remove the organic template, thus achieving the final ordered mesoporous material, with open pore structures like those of the liquid crystal mesophase [112]. This typical “bottom-up” synthesis has the nonpareil advantages of high flexibility and universal applicability [112]. Soft-templating approaches comprise three sub-pathways that have been experimented with for the successful engineering of ordered mesostructured materials. They include cooperative self-assembly (CSA), true liquid crystal templating (TLCT), and evaporation-induced self-assembly (EISA) [106].

Despite being time-consuming, costly, and lacking synthesis flexibility [112], the alternative hard-templating approach allows to bypass the need for careful control of several physical and chemical factors, required by the soft-templating method [106]. This strategy utilizes well-ordered mesoporous silica solid materials, with an interconnected pore structure as “hard templates” to create replicates [112]. They are steeped with selected precursors and then thermally or chemically treated to promote the formation of the desired phases [106]. Upon the achievement of the desired level of solidification, the template can be selectively removed using hydrofluoric acid (HF) or sodium hydroxide (NaOH) solutions. The final meso-structure is obtained as the negative replicate of the hard-template’s porous structure [106,112].

Cadmium sulphide (CdS), zinc sulphide (ZnS), cadmium selenide (CdSe), zinc selenide (ZnSe), cadmium zinc selenide alloy (CdS_x_Zn_1−x_Se), cadmium zinc sulphide alloy (CdS_x_Zn_1−x_S), copper sulphide (CuS), silver sulphide (Ag_2_S), polyoxometalate/silver sulphide/cadmium sulphide (POM/Ag_2_S/CdS), nickel disulfide (NiS_2_), iron disulfide (FeS_2_), cobalt di sulphide (CoS_2_), WS_2_, tungsten di-selenide (WSe_2_), MoS_2_, and molybdenum di-selenide (MoSe_2_)) using the hard-templating method [106]. The microstructure of MMCAGs prepared via the soft and hard template methods is provided in the Appendix A. Appendix A shows scanning electron microscopy (SEM) images illustrating the ordered mesoporous structures of samples obtained employing a soft-templating method, using surfactants (Appendix A) and a hard-templating method, using zein as a pore-forming agent (Appendix A). The images at ×45 and ×250 enlargements (from the left side) confirm the presence of macropores, while those at ×15,000 reveal the nanofibrous structure of the studied materials. Additionally, Appendix A shows the macro-porous structure of the samples analyzed using X-ray microtomography techniques. Based on these observations, it can be concluded that these methods enable the fabrication of AGs with a hierarchical porous structure. Collectively, the core–shell structures of alginate-based aerogels prepared using surfactants (Pluronic F-68 in this study) demonstrated macropores in the outer shell, while mesopores in the inner core part. The micro-CT results revealed macropore sizes in the range 16–323 μm, while the mesoporous structure of samples was characterized by a high SSA (657–673 m^2^/g) and a high specific mesopore volume (4.0–8.6 cm^3^/g).

The second method, using zein as a pore-forming agent, provided structures with unevenly distributed macropores in the range 5–195 μm in size, mesopore volume of 15.1–17.7 cm^3^/g, and a high SSA (592–640 m^2^/g).

The foaming method in a carbon dioxide medium afforded materials with macropores ranging from 20 to 3 mm, a SSA of 112–239 m^2^/g.

#### 4.1.2. Case Studies Concerning Some Relevant Applications of CAGs

CAGs have been experimented for the removal of heavy metal ions, and the results obtained by Nie et al. using K–Co–Mo–S_x_ (KCMS) CAGs as highly efficient sorbents are shown in Table 4 and Table 5 [149].

KCMS architecture encompasses Mo_2_^V^(S_2_)_6_ and Mo_3_^IV^S(S_6_)_2_ anion-like units with Co–S polyhedral structures, thus providing a Co–Mo–S covalent network that electrostatically attracts and hosts the K^+^ ions [149]. KCMS was extremely efficient in removing Ag^+^ (≈81.7%) and Pb^2+^ (≈99.5%) within five minutes, reaching >99.9% removal within an hour. KCMS displayed a notable removal capacity of 1378 mg g^−1^ for Ag^+^ and 1146 mg g^−1^ for Pb^2+^ [149]. The removal of Ag^+^ and Pb^2+^ from various water sources was also successful when highly concentrated and chemically diverse cations, including Hg^2+^, Ni^2+^, Cu^2+^, and Cd^2+^, anions, and organic species were present, [149]. By proper analytical techniques, it was demonstrated that the sorption of Pb^2+^, Ag^+^, and Hg^2+^ mainly occurred by the exchange of K^+^ and Co^2+^.

Additionally, the authors evaluated the practical use of the KCMS for wastewater treatment by analyzing its heavy metals removal efficiency from Mississippi River Water (MRW) and Tap Water (TW) [149]. These waters were appropriately further contaminated with Ag^+^, Hg^2+^, Pb^2+^, Ni^2+^, Cu^2+^, Cd^2+^, and Zn^2+^ metal ions at 10 ppm concentration each, achieving the overall concentration of 70 ppm for all seven metal ions [149]. Although other chemically diverse species, including cations, anions, and various organic species, were already present in both MRW and TW in high concentration, KCMS demonstrated a maintained removal efficiency for all added ions, as reported in Table 6 [149]. Among other ions, KCMS resulted particularly efficient in the removal of Ag^+^ and Pb^2+^ from MRW (>99%), lowering their final concentration under 1 ppb [149].

Raju et al. demonstrated the efficacy of antimony sulphide CAGs (SbS) in the selective sequestration of organic dyes, including rhodamine B (RhB), methylene blue (MB), and methyl violet (MV), from aqueous solutions, based on Lewis’s acid–base interactions [168]. As reported in Table 7 and Table 8, the adsorption kinetics were of pseudo-second order (PSO), and the equilibrium adsorption data were well explained by the Langmuir isotherm equation [168].

Starting from an initial concentration of 25 mg/100 mL of dyes in the tested solutions, 99% removal was realized in only 30 min treatment [168]. The SbS adsorption capacities, calculated using the Langmuir equation, were 442 (RhB), 303 (MB), and 210 mg/g (MV) [168], which were better than those of most of the adsorbents reported in the literature (Table 9) [168,169,170,171,172,173,174,175,176,177,178,179,180,181,182,183,184,185,186,187,188,189,190,191,192,193].

These findings evidence that CAGs could be excellent new ultralight materials for applications in the treatment of industrial effluents, paving the way for their future development as new platforms for the molecular filtration of hazardous dyes, derived from the discharge of industrial byproducts and other organic molecules. Metal chalcogenide ion-exchangers (MCIEs) have also shown great potential in removing radionuclides, thus helping in the remediate of the large amount of radioactive waste generated by the rapid development of nuclear energy [194]. Several research progresses on the removal of key radioactive ions (RI), including radioactive Cs^+^, Sr^2+^, UO_2_^2+^, lanthanide ions, and actinide ions by MCIEs, have been reported over the years. Table 10, Table 11 and Table 12 show the collection of the most relevant case studies concerning the removal of such radioactive ions by MCIEs containing alkali metal ions (Table 10), protonated organic amine cations (Table 11), as well as MCIEs prepared by the stepwise activation/intercalation method (Table 12).

MCIFs make the exchange of soft or relatively soft metal ions easy, due to their appropriately sized interlayer/channel/window spaces, their flexible open framework, and the strong affinity of the Lewis soft base S^2−^/Se^2−^ sites in their framework, thus demonstrating excellent selectivity and fast kinetic adsorption [194]. Unfortunately, despite Table 10 and Table 11 showing several MCAGs capable of sequestrating RI by ion exchange mechanisms, collectively, those capable of removing radioactive ions Cs^+^ and Sr^2+^ are limited, and several compounds that demonstrated positive results in laboratory settings failed in practical studies [194]. Therefore, to activate the ion-exchange properties of neutral MCAGs has become one of the key concerns in recent years [168]. In this regard, currently, the ion-exchange performance of metal chalcogenides has been enhanced by stepwise activation and cation intercalation activation methods [194]. Table 12 summarizes some case studies regarding the ion-exchange properties of MCIEs after improvement by the activation/intercalation method [194].

Several sulfide CAGs have been experimented with as sorbents for iodine (^129^I and ^131^I), which is a radionuclide, hazardous to humans and the environment, released in nuclear fuel reprocessing [225]. NiMoS_4_, CoMoS_4_, Sb_4_Sn_3_S_12_, Zn_2_Sn_2_S_6_, and K_0.16_CoS_x_ (*x* = 4–5) captured iodine (up to 225 mass%, 2.25 g/g of the final mass), thanks to strong chemical and physical iodine–sulfide interactions [225]. It was evidenced that Sb_4_Sn_3_S_12_ and Zn_2_Sn_2_S_6_ released captured iodine under thermal treatment at 150 °C, in the form of SnI_4_ and SbI_3_, respectively, which established the existence of chemisorption [225]. Conversely, NiMoS_4_, CoMoS_4_, and K_0.16_CoS_x_ released captured iodine in its elemental form, already at ∼75 °C, which is consistent with physisorption [225]. Preliminary investigations on consolidation of iodine-loaded Zn_2_Sn_2_S_6_ CAGs with Sb_2_S_3_, added as a glass-forming additive, produced glassy material whose iodine content was about 25 mass% [225]. The efficient capture of radionuclides with long half-lives such as technetium-99 (^99^Tc), uranium-238 (^238^U), and iodine-129 (^129^I) by nanostructured CAGs, such as Co_0.7_Bi_0.3_MoS_4_, Co_0.7_Cr_0.3_MoS_4_, Co_0.5_Ni_0.5_MoS_4_, PtGe_2_S_5_, and Sn_2_S_3_, thus preventing their transport into groundwater and/or release into the atmosphere, has also been reported by Riley et al. [140]. They showed the very efficient capturing of ionic forms of ^99^Tc and ^238^U, as well as nonradioactive gaseous iodine (i.e., a surrogate for ^129^I_2_ [140]. Collectively, PtGe_2_S_5_ demonstrated 98.0 and 99.4% removal efficiencies for ^99^Tc and ^238^U, respectively, and >99.0% for I_2_ (g). Minor capture efficiencies in the range 57.3–98.0% and 68.1–99.4% were observed for ^99^Tc and ^238^U, respectively, for different sorbents, while all CAGs were superior in the capture efficiency for iodine, showing >99.0% removal [140]. Later, the same authors reported the excellent capacity of tin sulfide (Sn_2_S_3_)-based CAGs of capturing iodine gas, due to Sn strong affinity for chemisorption of iodine to form SnI_4_ [142]. This study confirmed the utility of using GeS_2_ as a glass-forming additive. The addition of GeS_2_ to iodine-sorbed or iodine-free Sn_2_S_3_ CAGs caused better glass formation than Sn–S or Sn–S–I alone, and the quantity of iodine measured in the bulk glass of the consolidated iodine-sorbed Sn_2_S_3_ CAGs reached ∼45 mass%. Microwave sintering and hot isostatic pressing with iodine-sorbed Sn_2_S_3_ xerogels (XGs) were also experimented with to evaluate alternative consolidation techniques [142]. Riley et al. reported the performance of different types of SAGs and silver-modified SAGs, corresponding xerogels (XGs), and different types of MCAGs in the capture of gaseous I_2(g)_ and radionucleotides, observing the highest iodine loadings ever reported for inorganic sorbents and very good efficiency in managing radionucleotides for all materials [117]. Anyway, despite all the tested samples showing promise as next-generation adsorbents for active iodine and radionuclide remediation, MCAGs outperformed AGs and XGs in several cases [117]. Compared to silver-loaded aluminosilicates AGs and XGs, which demonstrated iodine capture values of 0.327–0.555 m _I_/m _S_, (intended as mass of iodine per mass of starting sample), MCAG demonstrated values of 1.60–2.40 m _I_/m _S_, establishing an iodine removal efficiency 2.9–7.3-fold higher than that of other AGs and XGs, which in turn already outperformed the efficiency of other inorganic sorbents [117]. Concerning the use of different MCAGs, including Co_0.7_Bi_0.3_MoS_4_ (CoBiMoS), Co_0.7_Cr_0.3_MoS_4_ (CoCrMoS), Co_0.5_Ni_0.5_MoS_4_ (CoNiMoS), PtGe_2_S_5_ (PtGeS), and Sn_2_S_3_ (SnS) for the capture of uranium—238 (i.e., ^238^UO_2_^2+^) and technetium—99 (i.e., ^99^TcO_4_^−^) ions from solution, removal efficiency values in the range 57–98% and 68–99% were observed for the absorption of ^99^Tc and ^238^U, respectively. These values were comparable or higher than those observed for other ion exchangers such as covalent organic frameworks (COFs) (95–99%) and metal–organic frameworks (MOFs) (81–99%) for ^99^Tc and comparable with those observed for magnesium oxide AGs (97–99%) for ^238^U [27].

#### 4.1.3. Author’s Summary and Considerations on CAGs

CAGs, intended as chalcogenide nanoparticle and cluster-based AGs (chalcogenide aerogels), are the latest material to arrive in the field of AGs and are now gaining notoriety due to properties not available in conventional AGs. Despite this, they are still too little studied, scarcely patented, experimented with only in laboratory settings, and not marketed. Following, I summarized the current research status, the main trends, and the challenges regarding the future research and directions for the development of CAGs.

##### Current Research Status and Trends of CAGs with Key Issues in Future Research and Directions for Their Future Development

CAGs have rapidly emerged as promising materials, due to their tunable optical band gap (from infrared to the visible range), highly polarizable surface, chemical activity without post-synthesis modification to add active functions, and adjustable structure. As reported by Ha et al., the current research status and trends concerning CAGs mainly regard the optimization of the synthesis of metal chalcogenide aerogels (MCAGs) and their experimentation, for environmental remediation and as catalysts and electrocatalysts for sustainable energy generation [106]. As reported in the previous tables, laboratory experimentations have proven that MCAGs possess great efficiency in the removal of several heavy metal ions from artificially contaminated water, by adsorption [149]. Concerning the removal of Pb^2+^ and Ag^2+^, MCAGs outperformed several other adsorbents experimented for the same scope [149,150,151,152,153,154,155,156,157,158,159,160,161,162,163,164,165,166,167]. Additionally, MCAGs have demonstrated strong efficiency (99% removal) in the adsorption of Pb^2+^ and Ag^2+^, also in practical settings, such as wastewater and water from the Mississippi River, in the presence of other cations, anions, and organic contaminants [149]. MCAGs have been reported as efficient adsorbers of dyes (99% removal in 30 min), including RhB, MB, and MW from water, via both adsorption and catalytic degradation mechanisms, with PSO kinetics [168], outperforming most of the adsorbents reported in the literature [168,169,170,171,172,173,174,175,176,177,178,179,180,181,182,183,184,185,186,187,188,189,190,191,192,193]. Moreover, MCAGs have shown great potential in the management of radionucleotides by ion exchange mechanism [194,195,196,197,198,199,200,201,202,203,204,205,206,207,208,209,210,211,212,213,214,215,216,217,218], which can be further improved by inserting in their structure protonated organic amine cations [226,227,228,229,230,231,232,233,234,235,236,237,238,239,240,241,242,243,244,245,246,247,248,249,250,251,252,253,254,255,256]. However, a key concern in recent years has been to succeed in activating the ion-exchange properties of neutral CAGs. Stepwise activation and cation intercalation activation methods, which work efficiently in improving the ion-exchange capacity of MCAGs, represent the current trend [219,220,221,222,223,224]. MCAGs have also demonstrated to outperform other AGs and XGs in the capture of iodine, demonstrating a removal efficiency higher by 2.9–7.3 times [117]. To promote the practical application of MCAGs, persistent efforts have been undertaken over the past two decades to introduce mesoscale porosity into their structure to form meso-porous metal chalcogenide (MMCAGs) materials [114]. The possession of meso-porosity is a pivotal characteristic that MCAGs should have to be more extensively developed and applied in a vast range of nanotechnological areas, including adsorption, catalysis, and energy conversion [114]. Additionally, the presence in MCAGs of mesoporous framework architectures, with higher and more accessible SSAs, can further enhance their mass-transport capacity. Also, meso-porosity allows the infiltration of selected species and can improve the reactivity of the semiconductive and catalytic surfaces of MCAGs, thus enabling the development of new functionalities [111,114,115,116]. To now, despite the great progress, the development of effective synthetic approaches to produce high-quality functional MMCAGs is still an ongoing challenge [111]. Particularly, research on the optimization of the synthesis of MMCAGs was and is focused on solving problems connected to the general instability of precursors and to the high intricacy of coordination, condensation, and polymerization chemistry of MCAGs. Due to these issues, the synthesis of MCAGs still presents great difficulties, especially if specific morphologies and functionalities are desired [112]. Currently, some of these obstacles have been surmounted by experimenting with new synthetic methods, which allow for controlling the precursor material interactions, also including those among the structural building blocks. Specifically, the preparation of MMCAGs with significant structural variations was achieved by two wide categories of synthetic approaches, including template-assisted and template-free strategies [112]. It is of paramount importance great efforts by researchers to further optimize the synthesis of MMCAGs and reduce their costs to obtain more advanced materials, scale up their production, and translate them to the market. In fact, it is noteworthy that MMCAGs offer valuable advantages for various surface or interface-associated functions, including adsorption, separation, and catalysis, which could meet the emerging environmental and energy-related demand [114,115,116,171,172]. A challenging drawback concerning MCAGs consists of their fragility and friability when used for environmental applications requiring high-flow gas streams or in extreme pH aqueous conditions. A current trend to address this issue consists of embedding the active sorbent in a passive matrix, providing chalcogenide PAN composites [141]. Despite this inactive matrix mass added that decreases the capacity of the overall composite, it increases its mechanical integrity [141]. Furthermore, the collected information on MCAGs leads to the conclusion that the most challenging problems concerning MCAGs, which strongly hamper their widespread and intensive experimentation, the upscaling of the current studies at the patent level, their translation into practical applications, and commercialization, are the very limited availability of MCAGs precursors, and the high costs of the commercially available ones.

### 4.2. Metal-Based Aerogels (MAGs)

As previously reported, the advances in material and technique development over the years, allowed the extension of AGs structures from the most conventional ones made of oxides [49,257] and polymers [258,259], to advanced architectures encompassing nanocarbons, nitrides, carbides, metal–organic scaffolds, semiconducting quantum dots and pure metal [260,261,262,263,264,265,266,267], thus enabling their AGs applications in catalysis, energy conversion, and storage, as well as environmental remediation. Furthermore, while numerous investigations exist concerning metal oxide AGs, those on metal-based AGs (MAGs) are in the early stages and are limited to laboratory experimentation. Recently, the nano-smelting of hybrid polymer–metal oxide AGs has provided materials including Fe, Co, Ni, Sn, and Cu. The method is based on the carbothermal reduction of polymer oxide composite AGs. Specifically, carbothermic reactions involve the reduction of substances, often metal oxides, using carbon as the reducing agent [268]. The reduction is usually conducted in an electric arc furnace or reverberatory furnace, depending on the metal ore used, using temperatures of several hundred degrees Celsius. A main application of carbothermal reduction consists of the iron ore smelting, following the reaction scheme below.2Fe_2_O_3_ + 3C → 4Fe + 3CO_2_

In the beginning, the source of carbon was charcoal and, later, coke. Charcoal is produced by burning wood under a limited oxygen supply. Highly porous carbon-AGs (C-AGs) are made by a similar method, via the pyrolysis under N_2_ or Ar of organic resorcinol-formaldehyde (RF) AGs. AGs with a low loading of <5% *w*/*w* iron have been explored in catalysis [268]. Importantly, pyrolysis of Fe^3+^-doped RF networks of nanoparticles yields Fe-doped C aerogels in one step [268]. The condensation of resorcinol (R) and formaldehyde (F) is catalyzed effectively by HCl. Leventis et al. engineered cast iron AGs by nano-smelting merged lattices of carbon and iron oxide AGs (*n*-RF-FeO_x_ and X-RF-FeO_x_, where n stands for native and X for cross-linked) at 800 °C in air [268]. Different pyrolysis temperatures from 200 to 1000 °C were explored to investigate the temperature effect on SSA of AGs [268]. However, a fundamental challenge, which remains unsolved, consists of the successful use of transition metals to produce MAGs with core–shell architectures [269]. In this context, Jiang et al. experimented with a one-step auto-programmed method to synthesize a core–shell Cu@Fe@Ni MAGs [269]. Electro activating (EA) Cu@Fe@Ni MAGs, the iron inner shell moves into the nickel outer shell, thus providing a novel catalyst (EA-Cu@Fe@Ni) which exhibited high catalytic activity and low oxygen evolution reaction (OER) overpotential (240 mV at 10 mA/cm^2^), which was much smaller than that of bimetallic CuNi (320 mV), CuFe (390 mV), and RuO_2_ (271 mV) AGs [269]. Raman measurements evidenced and confirmed that, during the EA process, the outer layer of EA-Cu@Fe@Ni was composed of NiOOH doped with iron, which resulted in the high OER performance. Among MAGs, noble metal aerogels (NMAGs) [270] are a new class of nanostructured materials [271,272] displaying high electrical conductivity, catalytic activity, and plasmonic features proper to noble metal NPs and the structural attributes of AGs. NMAGs can be applied as (electro)catalysts [273,274,275,276,277], nano enzymes [278,279], sensors [280,281,282,283], self-propulsors [284], and in plasmonic technologies [285,286,287]. They can be developed as mono-, bi-, and multi-metallic noble metal aerogels (NMAGs) consisting of Ag, Au, Pt, and Pd. The main synthetic methods to achieve NMAGs have been discussed in the previous section of this paper. The following examines several case studies on the strategies developed by researchers to optimize NMAGs’ preparation and their suggested applications.

#### 4.2.1. Sol–Gel Methods to Prepare Noble Metal and Metal Aerogels (NMAGs, MAGs) 

Figure 7 schematically summarizes the methods needed to prepare NMAGs and MAGs.

An appropriately modified sol–gel method is mainly used for producing NMAGs under soft conditions. Specifically, it consists of either a two-step or a one-step sol–gel procedure, both of which are completed by a supercritical CO_2_ drying step [37]. These methods have allowed the production of noble metal monolithic AGs, having high SSA and wide-open pores [37]. Due to these properties, monolithic NMAGs may be applied as sensors, as well as in heterogeneous gas phase catalysis and electrocatalysis [37,271]. The two methods mentioned above differ in the use or not of a separated NP colloidal solution [37,288]. While the two-step procedure starts with the reduction of selected metal ions to metal NPs (MNPs) followed by gelation, the one-step method consists of an immediate in situ spontaneous gelation process of metal ions [37,271,289]. In examples of the two-step strategy, different monometallic NPs (3–6 nm), covered with citrate, were synthesized by reduction of the metal precursors (HAuCl_4_, AgNO_3_, H_2_PtCl_6_, or PdCl_2_), with NaBH_4_, using trisodium citrate as a stabilizer [37,270,290]. Conversely, hollow preformed bimetallic nano-shell particles (NSPs) were prepared via galvanic displacement reaction between AgNPs stabilized with citrate and metal precursors (HAuCl_4_, K_2_PdCl_4_, and K_2_PtCl_4_) [291]. Also, thiolate-coated Ag NSPs can be obtained via rapid chemical reduction of preformed Ag_2_O NPs [292]. In all descripted cases, a secondary gelation step is needed, upon which the preformed metal NP solutions or their mixtures undergo condensation reactions [37]. Gelation can be induced by intentional destabilization via solution concentration by 10–50 times, using polystyrene centrifuge filters or employing a rotary evaporator, and minimizing or removing residual stabilizers and impurities, by water washings [37]. The gelation then takes place by leaving the concentrated NP solutions at room temperature or under thermal treatment (323–348 K) [37]. Conversely, destabilizers such as ethanol, H_2_O_2_, etc. can be added to promote the gelation of the concentrate NPs solution [37]. Unfortunately, by these destabilization methods, the formation of hydrogels is very long (about 1–4 weeks), which is an inconvenience that significantly increases production costs, thus greatly limiting their extensive applications [37,288]. Currently, several tactics have been developed to shorten the gelation time and to accelerate gel kinetics [37]. In fact, various novel destabilization methods have been proposed, including the modification of the synthetic parameters (temperature and disturbance), the addition of extra initiators (dopamine (DA), salts, tetranitromethane (C(NO_2_)_4_), and the use of NaBH_4_ [23,290,291,292,293,294,295,296] and hydrazine monohydrate (N_2_H_4_·H_2_O) [288]. Several monometallic hydrogels, including Au, Ag, Pt, and Pd, and multi-metallic ones, including Au–Ag, Au–Pd, Pt–Ag, Pd–Ag, Pt–Pd, Au–Ag–Pt, Au–Pt–Pd, Ag–Pt–Pd, and Au–Ag–Pt–Pd, have been prepared using this two-step approach, which were transformed in AGs by supercritical CO_2_ drying, previously described [270,288,290].

In the one-step strategy, gelation occurs spontaneously and simultaneously to the in-situ reduction of selected metal precursors with NaBH_4_ in a single step, avoiding the step previously necessary to perform adequately stabilized NPs [37,288]. Pd-protected α, β, γ-cyclodextrin (CD) (Pd_α, β, γ-_CD) hydrogels [297], pure Pd and Pt hydrogels, and bimetallic Pt_n_Pd_100–n_ hydrogels were prepared by this method [297,298]. Also, in this case, a long gelation time (3–10 days) is required, with the same negative consequences described above for the two-step strategy. Additionally, the large amounts of organic residuals (44 wt%) in the final product entails great difficulties in investigating the intrinsic activity of NMAGs. With the aim of addressing these issues, Zhu et al. proposed an in situ kinetically controlled reduction method and synthesized M–Cu (M = Pd, Pt, and Au) hydrogels within only 6 h by increasing the reaction temperature [299]. Furthermore, Shi et al. similarly fabricated, for the first time, AuPt_x_ bimetallic hydrogels at 60 °C in 2–4 h [300]. Regardless of the synthetic method used, supercritical CO_2_ drying, after the replacement of water in the pores of the hydrogels with acetone and further with liquid CO_2_, has ever demonstrated to be the most appropriate way to maintain the internal architecture of the hydrogel during the passage to the dry state of final AGs. Specifically, by using supercritical CO_2_ drying approaches, factors that may lead to the collapse of the fragile pores inside the structure are minimized, thus allowing the hydrogel to dry with very little shrinkage [37]. Also, CO_2_ supercritical drying allows to obtain AGs with higher SSA, intact pore appearance, and pore volume greater than that of AGs obtained using conventional drying methods [37].

Additional non-sol–gel synthesis techniques, including dealloying, combustion, and templating approaches, have been reported to prepare MAGS and NMAGs, which have been described in detail in the following Sections [289].

#### 4.2.2. Microstructure of MAGs by Sol–Gel Methods

In this Section, Appendix A, included in the Appendix A, are discussed according to the information provided by their authors. Numerical citations of papers from which Appendix A and related discussions present in this Section have been taken, have been reported in their captions respecting the numbers reported in the Appendix A description at the end of this paper, which followed the order of the quotations present in the main text. Specifications about the copyright license needed have also been included in the Figures’ captions. As mentioned above, readers can find the references of such citations in the reference list after the last reference quoted in the main text. Appendix A shows the FE-SEM images of α-Ni (OH)_2_ AGs prepared using a two-step sol–gel method, followed by a freeze-drying technique (Appendix A) and of the NiO/Ni AGs. The latter AGs, which demonstrated good porosity, were obtained by synthesizing α-Ni (OH)_2_ AGs upon annelation at 400 °C. The network-like continuous structures of such calcinated AGs looked made of nanoflakes of Ni (Appendix A). SEM micrographs of α-Ni (OH)_2_ AGs and calcinated NiO/Ni AGs demonstrated flaky nanoporous wafer-like architectures. While α-Ni (OH)_2_ samples demonstrated remarkably high aggregation, annelation provided NiO/Ni samples with more open structures and less aggregation. Collectively, the SSA values and porous nature of such AGs, investigated by conducting nitrogen sorption tests, were similar, providing 54.8 m^2^/g and 55.6 m^2^/g, respectively. The volume of adsorption was also noted to be similar, and evidenced mesopores and macropores with most pore sizes ranging from 10 to 50 nm, with some macro-porous regions with pores>50 nm in size. Conversely, Appendix A schematizes the one-step hydrothermal self-assembly method used to prepare a platinum NPs supported graphene AG (Pt/3DGA) catalyst (Appendix A) and the SEM images of the Pt/3DGA at different magnifications (Appendix A). The structure of the AG was stabilized by the simple one-step method, which reduced production costs compared to the freeze-drying technology previously described, but also optimized the loading method of NPs. The produced Pt/3DGA catalyst had an extremely low weight and exhibited a relatively unfastened porous structure. SEM micrographs evidenced that Pt/3DGA possessed an interconnected 3D porous architecture, with pores in the range 2–10 µm (Appendix A). The Pt/3DGA AG manifested to be highly cross-linked, with large SSA, and to possess high dispersion capacity, as well as good electrical conductivity. Specifically, analytical data for Pt/3DGA revealed a BET SSA of 227.89 m^2^/g pore size of 13.65 nm, and a pore volume of 0.279 cm^3^/g.

#### 4.2.3. Non-Sol–Gel Methods to Prepare Metallic Aerogels

The synthetic procedures different from sol–gel methods needed to achieve metal nanofoams (MNFs) and MAGs include dealloying, combustion, and templating approaches. Thanks to these syntheses, some remaining challenges plaguing the MAGs and NMAGs production have been relatively solved [37]. Specifically, the too long synthesis time and the difficulty in the control of AGs shape, which experts frequently encounter when one- and two-step methods are used, have been met. Additionally, the bio-templating strategy has enabled a robust and efficient control on the nanostructures (NSs) diameter length and on the final shape of the monolith [37]. Also, the salt templating approach is an attractive up-scalable synthesis, which provides MAGs and NMAGs with high SSA and porosity [37,289]. Specifically, dealloying (DEA) is a process of corrosion that eliminates in a selective manner one or more metals from alloys, allowing the residual metal to transform voluntarily into porous structures. Upon its discovery, DEA was adopted to achieve films, foams, and porous NPs [301,302,303,304,305,306]. Alloys made up of Ag and Au were used in dealloying processes during which silver was removed by dissolution, while Au atoms aggregated, forming a relatively homogeneous Au-based 3D porous structure. Unfortunately, the limited number of alloy combinations available and the difficulty of producing materials with particle size ≤ 1 mm represent the most concerns associated with DEA [37,307]. Anyway, also when materials with particle size of 10 nm were achieved, they demonstrated a very low SSA < 10 m^2^/g, thus making DEA a low attractive approach [37]. In place of DEA, Au and Pd AGs were produced by combustion methods (COMs). COM starts with the burning of metal complexes by highly energetic ligands [307,308], providing Au and Pd monolithic AGs with SSA of 10.9 m^2^/g and 36.5 m^2^/g, respectively, suggesting the existence of the same disadvantages described for DEAs [37]. Conversely, bio-templating (BTEM) is a tactic that uses material chemistry to obtain materials with a precise biological morphology, derived from the selected bio-template, but with different compositions and shapes [37]. BTEM makes use of biological molecular structures, including microorganisms, viruses, and biomolecules (DNA, cellulose, and proteins) as template models. Bio-templated NSs can be achieved by different methods, such as binding preformed NPs, vapor deposition, or electroless deposition of metals onto the template surface [37,309,310]. Anyway, despite this approach having allowed to prepare several nanomaterials and porous films, it remains in its infancy for the preparation of NMAGs [37]. Salt-templating (STEM) approach is instead an alluring tactic, characterized by easy scaling-up, short time synthesis, and low-cost production, which can provide noble metal NFs and NMAGs as thin films, with potential to be applied as electrodes [37,302,311]. Burpo et al. used non-soluble salt needles as templates for engineering porous macro-tubes and macro-beams with tunable densities, which assembled into arbitrary shapes, including thin films [309,310,311,312]. Salts generated by mixing varied concentrations of [PtCl_4_]^2−^ and [Pt (NH_3_)_4_]^2+^ ions are known as Magnus’ salts [289]. They were reduced and used as non-soluble precursor salt needles templates to prepare monolithic NMAGs [37,312]. Highly porous macro-tubes, with shapes in agreement with the salt templates used as models, were achieved. X-ray photoelectron spectroscopy (XPS) and X-ray diffractometry (XRD) analyses showed the presence of Pt structures without the presence of oxide, while electrochemical impedance spectroscopy (EIS) indicated a specific capacitance of 18.5 F/g and an electrochemically active surface area (ECSA) of 61.7 m^2^/g. Cyclic voltammetry (CV) showed peculiar hydrogen adsorption and desorption and Pt oxidation–reduction peaks [312]. SEM micrographs evidenced porous sidewalls consisting of Pt NPs (~100 nm in diameter), in turn encompassing textured nanofibrils with a 4.9 ± 0.7 nm average diameter [312]. Reduction was carried out with 0.1 M NaBH_4_ at 1:50 (*v*/*v*) salt needles. It is noteworthy that two [Pt (NH_3_)_4_]^2+^ are needed in solution to conserve a charge balance for each [PtCl_4_]^2−^ ion reduced [37]. By combining different square planar noble metal salts to form Magnus’ salt derivative needle templates, bimetallic and alloy NMAGs have been engineered [37]. Burpo et al. investigated the salt-template reduction–dissolution path for the formation of Pt@Pd hierarchical metal NSs [310]. Particularly, the addition of [Pt (NH_3_)_4_]^2+^ ion solutions to different concentrations of [PtCl_4_]^2−^ and [PdCl_4_]^2−^ anions yielded bimetallic salt templates with lengths from 15 to 300 µm [310]. Templates were chemically reduced with NaBH_4_, thus shaping Pt-Pd macro-beams with high porosity and square cross sections in compliance with the length of the initial salt template [310]. Each macro-beam structured AG showed porous sidewalls with internal primary Pt and Pd NPs or fibrils having diameters of 8–16 and 4–7 nm, depending on the Pt/Pd ions ratio present in the original salt template. ECSA values were 23.2–26.7 m^2^/g. It is noteworthy that these values were almost half those of SSA measured by BET analysis of Pt@Pd bimetallic AGs synthesized by Bigall et al. using the two-step gelation approach [270]. Anyway, in the salt template method, the salt template formed immediately, and its electrochemical reduction was extremely rapid [37].

##### Microstructure of MAGs by Non-Sol–Gel Methods

In this Section, Appendix A, are discussed. Numerical citations of papers from which Appendix A and related discussions presented in this section have been taken have been reported in their captions, respecting the numbers reported in the SMs description at the end of this work, which in turn followed the order of the quotations present in the main text. Specifications about the copyright license needed have also been included in the figure’s captions. As mentioned above, readers can find the references of such citations in the reference list after the last reference quoted in the main text.

##### Dealloying Method

Appendix A schematizes a process used to prepare cobalt porous gold NPs (CP@Au@NPs), including a dealloying phase (b) (Appendix A) and the secondary electrons SEM image (SESEM) of the porous gold nanoparticles (P@Au@NPs), obtained by dealloying the amorphous precursor (Appendix A). The morphology of freshly produced de-alloyed P@Au@NPs displayed micrometric and irregularly shaped islands of pure gold (Au), pervaded by a well-defined porous structure. Pores demonstrated dimensions <1 µm.

##### Combustion Method

The combusting method was used to transform the MnO_2_/bacteria (BMB) and the MnO_2_/bacteria/Ni (BMB-Ni) porous composites, prepared by a bio-templated method based on *Pseudomonas putida* cell surface display technology, to obtain the biogenetic MnO/C (CMB) B800 and the MnO/C/NiO (CMB-Ni) porous composites. The SEM images of these materials are exhibited in Appendix A, respectively. Combustion was carried out in a tube furnace at 800 °C under argon atmosphere. Upon decomposition of the organic carbons of the organisms, during combustion process, a reducing atmosphere was generated, which reduced MnO_2_ to MnO. In the interim, the residual inorganic carbon was transformed into a carbon coating and an electroconductive carbon matrix, while cavities and macropores took the place of organic matter. The pore size distribution curve of the samples under calcination conditions of 800 °C revealed pores in size 14.3 and 98.4 nm for B800 (CMB) and of 11.9 and 104 nm for CMB-Ni.

##### Bio Templating Method

Appendix A illustrates the SEM images of biogenic MnO_2_/bacteria (BMB) porous composites above-mentioned, prepared by a bio-templated method based on *Pseudomonas putida* MB285 cell-surface. The aggregates looked like unbalanced spherical secondary particles, encompassing exfoliated MnO_2_ fixed to bacteria surfaces and dispersed spherical MnO_2_, as well as showed a relatively uniform size of 20–30 µm. Their structure appeared dense and preserved intact the morphology even at a SEM operational voltage of 200 kv. The BMB aggregates had a relatively uniform size with a diameter of 20–30 µm. The analysis of the BET specific area and porosity indicated that the SSA of BMB was 20.935 m^2^/g, while the Barrett–Joyner–Halenda (BJH) pore size distribution was in the range 20–33.97 nm, which is classed as a mesoporous material.

##### Salts Templating Methods

Appendix A shows the SEM images of biomass-derived porous carbon materials, synthesized by a molten chloride salt templating technique and successive KOH activation (MHPC-700, MHPC-800, and MHPC-900) (Appendix A). These materials demonstrated a good balance between high SSA and mesopore volume. For comparison purposes, Appendix A shows the SEM micrograph of the NHPC-700 pre-carbonized in nitrogen atmosphere without molten salt. While a thick carbon block architecture prevailed in NHPC-700 morphology, a sheet-like structure dominated in MHPC-700 morphology, establishing that the latter structure formed only during the pre-carbonization process in molten salt. The molten salt presence promoted both the generation of mesopores and served to provide the active sites for the KOH activation, which allowed the formation of micropores in the final carbon material. Collectively, the mesopore–micropore structure of the porous carbon final materials can be tuned by changing the pre-carbonization temperature. Pore textural properties of the as-obtained porous carbon materials consisted of a total BET SSA of 524. 361 and 309 cm^3^/g, a total pore volume of 0.42, 0.28 and 0.27 cm^3^/g, a micropore volume of 0.19, 0.15, and 0.21 cm^3^/g and a mesopore volume of 0.23, 0.14, and 0.15 cm^3^/g for MHPC700, 800, and 900, respectively.

#### 4.2.4. Case Studies on Synthesis and Applications of NMAGs and MAGs

The sol–gel approach establishes extraordinary advantages for the versatile synthesis of nanosized NMAGs under mild conditions and constitutes a powerful “bottom-up” tactic for creating nanocomposites that meet the most difficulties associated with the synthesis of AGs [271]. In this regard, the one-step method has allowed for combining the NPs preparation and the gel formation in a single step, thus resulting in a more convenient and rapid production than the two-step procedure. Despite this, a too long gelation time persisted (3–10 days) and more than 44 wt% of organic matter resided in the final product, thus making a reliable test of the intrinsic activity of NMAGs difficult. In addition, still indefinable formation mechanisms hamper an efficient structure/composition manipulation, thus impeding the on-demand design of NMAGs, for a wider range of practical applications. Wen et al. decreased the gelation time of palladium (Pd)-based NMAGs to less than 5 min by the electrostatic cross-linking between cation Ca^2+^ and the citrate-covered Pd NPs. The as-prepared Pd AGs displayed high SSAs of 40–108 m^2^g^−1^ [313] and demonstrated high performance in the bio electrocatalytic oxidation of glucose [313].

Highly adjustable NMAGs were manufactured in 2–120 h by Du et al., by activating specific ion effects (ASIEs) [314]. ASIEs are primarily allowed to bypass limitations of gelation methods, which hinder the on-demand production of AGs for precise applications. Moreover, this method permits different single/alloy AGs with adaptable composition (Au, Ag, Pd, and Pt), ligament sizes (3.1–142.0 nm), and unparalleled nanosized core–shell morphologies [314]. Such NMAGs were superior in the development of programmable self-propulsion tools and in electrocatalytic alcohol oxidation reactions [314]. Later, Du et al. developed NMAGs containing Au, Ag, Pd, Pt, Ru, Rh, and Os, within 6–12 h, by the innovative excessive-reductant-directed gelation strategy [315]. This approach successfully used the ligand chemistry for preparing Au AGs. Materials with higher SSA (59.8 m^2^ g^−1^) were achieved, and the possible composition of NMAGs was enlarged to almost all the most common noble metals [315]. Authors demonstrated that such AGs owned impressive electrocatalytic performance for the ethanol oxidation reactions and OERs [315]. Furthermore, an unconventional organic-ligand-enhancing effect was observed, thus paving the way for perceiving efficient electrocatalysts for broad material systems [315]. Wang et al. reported the controlled preparation of Au@Pt bimetallic AGs (BIAGs) in segregated, alloy, and core–shell typical structural configurations [316]. BIAGs exhibited enhanced peroxidase- and glucose oxidase-like catalytic performances compared to their monometallic counterparts [316]. Au@Pt_x_ BIAGs were similarly fabricated by Shi et al., at 60 °C in 2–4 h, which demonstrated improved electrocatalytic performances [317].

With the aim of accelerating hydrogel formation, improving the gelation kinetics, and simplifying the synthetic procedure, Zhu et al. engineered a collection of M@Cu (M = Pd, Pt, and Au) BIAGs by the in-situ reduction of metal precursors, at elevated temperature. Particularly, Pd@Cu AGs showed an ultrathin nanowire grid and exhibited excellent electrocatalytic capacity in the ethanol oxidation reactions, thus being promising in fuel-cell applications. [318]. Wang et al. shortened further the gelation time to 1 h and adjusted the electronic structure of Pd@Cu AGs by using ionic liquids (ILs), thus optimizing their intrinsic activity [319]. Direct methanol fuel cells (DMFCs), based on IL/Pd_3_@Cu_1_ anode catalysts, revealed a power density higher than those containing only Pd_3_@Cu_1_ and commercial Pd catalysts, thus establishing the advantages of using these NMAGs for advanced electrocatalysis [319]. Other various alloy NMAGs, including Pt@Ni, Ir@Cu, Au@Cu, and Pt@Ru@Cu, were reported over the years [320,321,322,323]. Ir_x_@Cu AGs performed well in the catalysis of OERs [320], ternary Pt@Ru@Cu AGs showed potential for enhanced methanol electrooxidation [321], ultrafine Pd-anchoredAu_2_@Cu AGs boosted ethanol electrooxidation [322], while unsupported Pt@Ni AGs demonstrated enhanced high current performance and robustness, thus being applicable in fuel cell cathodes [323]. Also, Du et al. strongly accelerated the gel kinetics by applying an external stirring force. Specifically, performing a one-step gelation process, using NaBH_4_ as a reducing agent, various monolithic AGs were achieved in a short time (1–10 min), by exploiting the self-healing capacities of gel pieces [324]. Highly efficient photo-electrocatalytic activity was experimented for the as prepared AGs [324].

However, the use of NPs colloidal solutions remains the most noteworthy tactic for enriching the structural diversity of NMAGs. Au colloidal NPs solutions were prepared for the first time by Fan et al. via a laser synthesis and processing (LSPC) method [325]. Then, by utilizing a NaBH_4_-induced gelation, clean Au AGs possessing enhanced electrocatalytic performance, were formed in 8 h, without recovering to organic ligands [325].

Burpo et al. achieved controlled NMAGs macroscopic shapes and nanostructures, using the bio-templating technique [326,327]. To this end, both gelatine and cellulose nanofiber (Gel and CNF) hydrogels were used by authors as bio-templates to manufacture composite Pd@AGs (Gel-Pd@AGs and CNF-Pd@AGs) [326,327,328]. Specifically, CNF-Pd@AGs were prepared by exposing CNF hydrogels to Na_2_PdCl_4_ and Pd (NH_3_)_4_Cl_2_ solutions (1–1000 mM), followed by a reduction reaction with 2 M NaBH_4_ [327]. The process was accomplished by ethanol solvent exchange and supercritical CO_2_ drying, which allowed to maintain the original morphology of the hydrogel in the final AGs. Figure 8(1,2) illustrates the process used by Burpo et al. to manufacture CNF-Pd@AGs (1) and photos of the intermediates and final NMAGs (2).

Pd@AGs prepared using 1 mM and 10 mM Pd solutions showed 3D nanowire structures, with an average diameter of 12.6 ± 2.2 and 12.4 ± 2.0 nm, as well as a pore size of 32.4 ± 13.3 and 32.2 ± 10.4 nm, respectively. On the contrary, Pd@AGs engineered utilizing 50, 100, 500, and 1000 mM Pd solutions, displayed mean NPs diameters of 19.5 ± 5.0 nm, 41.9 ± 10.0 nm, 45.6 ± 14.6 nm, and 59.0 ± 16.4 nm, respectively [327]. CNF-Pd@AGs possessed SSAs, which decreased with increasing concentration of Pd salt and metal content. Conversely, pore size distribution depended on the template shape and noble metal concentration. Robustness of these AGs was higher than that of monolith counterparts containing only metals, thus resisting without breaking to repeated dropping and squeezing with tweezers. Furthermore, Cai et al. produced several hierarchically porous nanostructures by assembling pre-engineered NPs. Pd@Ni or Ni-Pd@Pt hollow nano-building blocks templates were first prepared with Ni NPs, via a galvanic replacement reaction (GRR). Subsequently, their gelation was promoted by increasing the temperature to 348 K for 6 h, thus achieving materials that can find applications in enhanced electrocatalysis [294,295].

Core–shell Pt-Pd_x_@Au AGs, encompassing an ultrathin Pt shell and a core with tuneable Pd_x_@Au alloy composition, were developed by Cai et al. [329]. The versatility of this strategy assured the extension of core compositions to Pd transition-metal alloys. The as-prepared Pt-Pd_x_@Au AGs exhibited strongly improved Pt utilization efficiencies in the oxygen reduction reactions (ORRs) [329]. The method proposed by Cai et al. could be a new possible strategy for the design of future core–shell electrocatalysts [329].

As previously reported, Burpo et al. synthesized Au@Cu NFs and Au@Cu@Pd macro-beams, following the salt-templating method [311]. Specifically, the oppositely charged [AuCl_4_]^−^ and [Cu (NH_3_)_4_]^2+^, as well as [AuCl_4_]^−^, [Cu (NH_3_)_4_]^2+^and [Pd (NH_3_)_4_]^2+^ square planar ion complexes, were combined, yielding short- and long-range ordered salt needles. The reduction of the Au@Cu salt solutions provided NFs, while that of Au@Cu@Pd salt solutions gave macro-beams like Magnus’ salt-templated platinum macro-beams [312]. Also, using [Cu (NH_3_)_4_]^2+^, [Pt (NH_3_)_4_]^2+^ and [PtCl_4_]^2−^ ions, platinum-copper salt needles were synthesized, which, upon reduction, formed Pt@Cu bimetallic macro-tubes [309]. Despite Au, Pd, and Pt nanostructures with smaller size being furnished by the one- or two-step gelation methods, Au, Pd, and Pt AGs developed by following the salt-templating method were achieved in a more rapid gel formation time, which took advantage of high relative local ion concentrations, leading to NP coalescence. Collectively, electrochemical impedance spectroscopy experiments and cyclic voltammetry demonstrated that all NMAGs developed by Burpo et al. via salt templating approaches possessed highly capacitive properties with electrochemically active SSA high up to 52.5 F/g [309,311,312]. Therefore, such NMAGs are suitable for catalysis, sensors, fuel cells construction, and energy storage applications. Using the information contained in a review by Wang et al., the following Table 13 was personally elaborated, which collects the main characteristics of the most relevant NMAGs developed so far [288].

##### Case Studies on Synthesis and Applications of Other MAGs

The most used reducing agent in both the one-step and the two-step sol–gel methods is NaBH_4_ [315]. Anyway, sodium hypophosphite (NaH_2_PO_2_) [343], ascorbic acid (AA) [344,345], dimethylamine borane (DMAB) [339], carbon monoxide (CO) [340,342], and glyoxylic acid monohydrate (CHOCOOH) [340] were proposed as alternative reductive compounds, especially in the one-step synthesis of NMAGs. Hope-Weeks et al. reduced CuO [346], CuO@NiO [347], or ZnO@CuO [348] to Cu_2_O and metallic Cu [349], in a 5% hydrogen and 95% nitrogen atmosphere, and used the unconventional CuBr_2_ to achieve the hydrogels via EA in dimethylformamide (DMF). After calcination, Cu AGs, having enhanced mechanical properties compared to those of AGs derived from CuCl_2_, were achieved. Zhang et al. obtained pure metallic copper AGs (Cu AGs) after annealing under a reducing atmosphere [350,351,352]. More recently, a new way to manufacture pure metallic iron AGs (Fe AGs) was proposed [353,354]. To this end, iron precursors were gelled using a poly-benzoxazine network as a template. Porous iron structure with 7% relative bulk density was achieved after carbonization and removal of the carbon network, which is applicable as energetic materials [353,354]. In this regard, Rewatcar et al. developed a new route to metallic AGs via carbothermal synthesis of monolithic Co (0) AGs from compressed cobalt-based xerogel powders cross-linked with polyurea [355]. Residual carbon was removed, providing carbon-free samples by high-temperature treatment of as-prepared Co (0) AGs under a flowing stream of H_2_O/H_2_ to prevent oxidation of the Co (0) network. Co (0) AGs demonstrated durability under harsh processing conditions in their application as thermites. When Co (0) was burned to CoO, the temperature exceeded 1500 °C, and the heat released (−55_2_ ± 2 kcal mol^−1^) was near both the theoretical value (−58.47 kcal mol^−1^) and that from well-known pressed-pellet iron/perchlorate thermites (−66.6 kcal mol^−1^) [355]. Moreover, Leventis et al. reacted the acidic hydrated CuCl_2_ solution with a network of CuO nanoparticles to induce one-pot co-gelation of a nanostructured network of a resorcinol-formaldehyde resin acting as the fuel [356]. Upon drying of the resulting wet gels to AGs and pyrolysis under Ar, the interpenetrating CuO/RF network undergoes a smelting reaction toward metallic Cu. Upon ignition in the open air, the as-obtained AGs sustained combustion, burning completely, leaving only a solid residue of CuO, which acted as redox mediator through the smelting reaction. Such materials are very promising as energetic materials for potential use as explosives, propellants, and pyrotechnics [356]. The same author engineered monolithic nanoporous iron materials via carbothermal reduction of interpenetrating networks of poly benzoxazine and iron oxide nanoparticles. Excess carbon was removed at 600 °C in the air, and oxides produced from partial oxidation of the Fe (0) network were reduced back to Fe (0) with H_2_ at different temperatures (300–1300 °C). Carbon-free and oxide-free Fe monoliths were infiltrated with perchlorates, dried exhaustively, and were ignited with a flame in open air. Depending on temperature, monoliths fizzled out (≤400 °C), exploded violently (500–900 °C), or behaved as thermites (≥950 °C) [357]. Also, Zhou et al. prepared metallic titanium AGs (Ti AGs) by using TiO_2_ AGs as a template and magnesium-thermic reduction with HCl [358]. Using the filiform M13 viruses, which are broadly used as bio-templates to construct several functional structures, Jung et al. constructed noble metal ruthenium (Ru) AGs (Ru AGs), respecting the M13 shape anisotropy, reasonable aspect ratio (length to diameter of ≈130), and low density. Freestanding bulk 3D M13@Ru AGs were achieved, which showed 10 nm and 14 nm nanowires, ultralight porous structures, an interconnected virus network, sheet-like structures, large tens of micrometres, and densities in the 4–10 mg cm^−3^ range [359]. Pores ranged from several to tens of micrometres, and Ru AGs with 10 mg cm^−3^ density displayed an elastic modulus of 2 kPa [359]. Furthermore, since the genome of M13 can be strategically designed to have capsid proteins capable of exactly binding numerous inorganic materials, Jung et al. prepared multi-component CoFe_2_O_4_ AGs made from inorganic complexed M13 structures, with adaptable functionalities (M13@CoFe_2_O_4_) [359]. Specifically, upon solvent evaporation, dilute suspensions of M13 virus furnished hydrogel 3D networks via van der Waals forces, which resulted in freestanding AGs with good shape retention, after freeze drying. Both M13@Ru AGs and M13@CoFe_2_O_4_ AGs demonstrated excellent mechanical properties with elastic behaviour up to 90% compression. By this advanced method, a vast arsenal of freestanding IAGs applicable as bio-scaffolds, for energy storage, in thermo-electrics, catalysis, hydrogen storage, etc., can be prepared [359]. The M13@Ru AGs were experimented with as Li-air battery electrodes and were tested both as doctor-bladed thin film and AG Li-O_2_ cathodes. As demonstrated by the first discharge–charge cycle (820 mAh g^−1^ vs. 60 mAh g^−1^), the M13@Ru AGs proved an improved performance in the ORRs and OERs, with respect to conventional doctor blade electrodes [359]. The macro-pores of M13@Ru AGs, ranging in size from 10 to 20 μm, provided increased effective SSA for the above-mentioned reactions, higher mass-specific performance parameters, and electrical conductivity [359]. Polyamide AGs with ferrocene as a monomer repeat unit were prepared in one step from ferrocene dicarboxylic acid and tris (4-isocyanatophenyl) methane [360]. Pyrolysis at ≥800 °C of ferrocene-based polyamide AGs gave monolithic carbon aerogels bearing Fe (0) nanoparticles of about 50 nm in diameter dispersed throughout their volume. These materials were trans-metalated with selected metal ions, in aqueous medium, replacing Fe (0) nanoparticles with Au, Pt, Pd, Ni, and Rh [360]. These materials demonstrated nanoparticles of 10–20 nm. All metal-doped carbons were monolithic and over 85% porous. Carbon-supported Au or Pt demonstrated catalytic activity in the oxidation of benzyl alcohol to benzaldehyde, carbon-supported Fe catalysed the reduction of nitrobenzene by hydrazine to aniline, while carbon-supported Pd catalysed two Heck coupling reactions of iodo-benzene with styrene or butyl acrylate [360]. The overall merit of those catalysts consisted of their immediate reusability, thus bypassing less efficient recovery processes like filtration.

## 5. Interdisciplinarity of Aerogels

AGs technology represents an interdisciplinary innovation skill, connecting inorganic and organic chemistry, supramolecular chemistry, the science of materials, environmental engineering, biotechnology, etc., and all science sectors, which can contribute to expanding the utility of these unique materials beyond traditional uses. Table 14 reports the main sectors of application of AGs with motivations and possible uses and/or deriving advantages.

In biomedicine, AGs have demonstrated great potential as controlled drug delivery systems, due to their tuneable pore sizes and large SSA, as well as the capacity to control release rates of pharmaceutical principles. SBAGs are under study for targeted cancer therapies, aiming to limit systemic side effects and to optimize therapy efficiency by delivering drugs directly to tumour sites [361]. In the energy sector, AGs are used to manufacture advanced components for batteries and supercapacitors. AGs relevance in this field is based mainly on their high SSA, which ameliorates the interactions between electrolytes and electrodes, thus improving the charge storage efficiency of devices for this use and the potential of electric vehicles and portable electronics [81,362]. Also, as dielectric materials, and due to the linearity existing between their density scales dielectric constant. NASA developed hydrophobic polyimide AGs to be used as patch antennas [365]. Moreover, pollution could strongly damage high-density integrated circuits utilized in military systems and space-based microelectronics [37]. In this context, nowadays available AG mesh contamination collectors (AMCCs) are capable of removing molecular and particle impurities possibly contained in a spaceship [366]. Moreover, in aerospace engineering, AGs are used as insulation materials to protect spacecraft and satellites against the severe conditions of space, mostly due to their insulation properties and resistance to extreme temperatures [363]. In environmental engineering, AGs are employed to construct filtration systems for air and water purification, due to their nanostructured networks, which efficiently can entrap fine particulates and contaminants, thus addressing the issue of environmental pollution [364]. The interdisciplinary research aims not only at optimizing existing applications, but also at finding new AG applications, enhancing their integration into commercial technologies. In fact, it is reported that an artificial muscle was engineered using carbon nanotubes (CNTs)-based AGs, which showed to be capable of exerting great force and transporting objects, one thousand times faster than a genuine muscle at an operating temperature of −196–1538 °C. [367]. Fabrics, paints, varnishes, and boats benefit from the addition of superhydrophobic AGs, which provide surfaces incredibly water-repellent, thus functioning as “self-cleaning” materials [366]. The introduction of alginate-lignin, nanocellulose, and chitosan reinforced with ceramic quartz fibre into SAGs represents a valuable strategy to achieve SAGs with robust frameworks suitable for enzyme encapsulation, thus promoting their practical applicability [37]. SAGs reinforced with fibres were also used as corrosion-resistant insulators, filtering media, for electrically charged constituents separators, shock absorbers, and even scaffolds for tissue engineering [368]. Carbon AGs having nitrogen groups on the surface demonstrated augmented capacity for capturing CO_2_, while polysaccharide-derived carbon AGs served as a valuable substrate for entrapping transition MCAGs (MoSe_2_), to improve the electrocatalytic activity in hydrogen evolution-based energy generation [369].

### Current Research Hotspots and Future Development Trends: Author Consideration

Collectively, the hotspots in the current research on AGs are mostly converged on the optimization of their synthesis and on the control of their mechanical capacity. On the other hand, thermal and acoustic insulation, pollutant adsorption, and the construction of Cherenkov detector radiators dominate the scene of their possible or actual applications. Anyway, synthetic methods and specifically sol–gel processes, are the most popular research topics, followed by the research finalized on regulating AGs mechanical properties. Obtaining AGs with suitable characteristics of robustness, durability, and resistance in different conditions, including the most prohibitive ones, strongly affects their possible and efficient application as thermal insulators, absorbents, and to engineer Cherenkov detector radiators. Minor flashpoint applications can include the use of AGs as vectors for catalysts, drug delivery systems, and optical devices. The intensive study on AGs has currently addressed several challenging aspects, thus allowing to hypothesize an expanded number of their possible applications in several sectors, not imagined so far. Anyway, many other challenges and opportunities for these materials will be foreseeable soon. In my opinion, future development trends should pay more emphasis mostly on the AGs eco-sustainable synthesis, to limit environmental pollution and to guarantee healthier procedures to operators. In this context, most consideration should be required for the possible preparation of AGs using green and pollution-free reagents, including rice husk ash, as a non-toxic source of silicon necessary to engineer SAGs. Similarly, synthetic methods using non-toxic solvents in the preparation and aging stages need to be better explored.

Other bearing conditions, including twist and bend stress ones, should be experimented with to further enhance the hardness and compressibility response of AGs.

Three-dimensional printing and computerized simulation techniques have allowed scientists in different fields to create structurally different materials with unparalleled properties and could also be applied to project an innovative structure of AGs. They can simulate the total distortion capacity of products with different structures, thus allowing experts to distinguish the structure with superior mechanical properties and deformation ability.

Desired 3D objects can be manufactured by 3D printing computer-aided design models. Anyway, these printed mass objects require proper solidification to maintain the original shape after 3D printing. In this regard, solidification techniques are still underdeveloped and need substantial improvements, mainly in terms of thermal solidification and freeze casting. Finally, several concerns remain regarding the unsolved challenge of scaling up the production methods. In this context, the rational combination of freeze casting with more intelligent temperature control systems could be an appealing tactic.

## Figures and Tables

**Figure 1 ijms-26-10696-f001:**
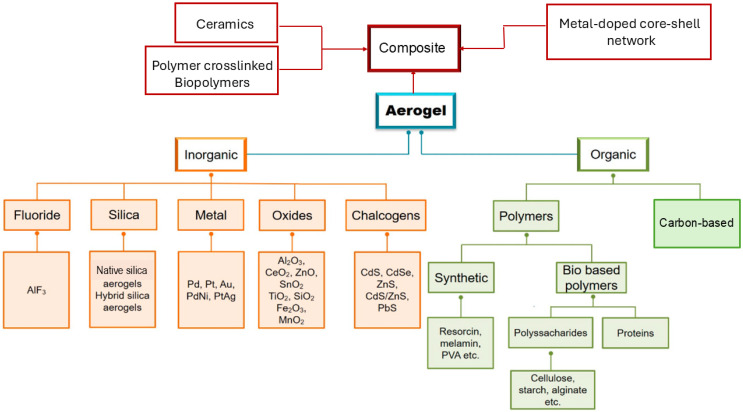
Classification of AGs according to the nature of the materials used as precursors. The scheme has been created by the author.

**Figure 2 ijms-26-10696-f002:**
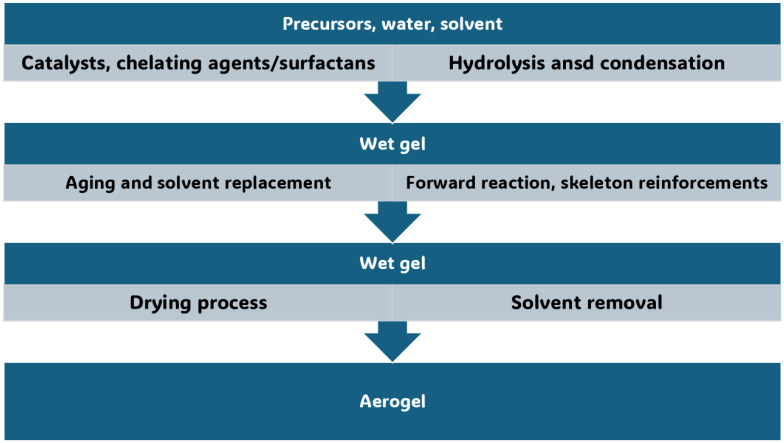
Sol–gel process to fabricate metal oxide AGs. The image is a production of the author.

**Figure 3 ijms-26-10696-f003:**
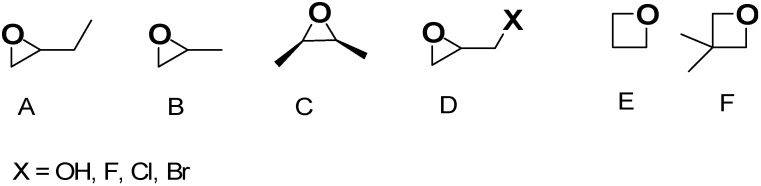
Epoxy molecules have mostly been experimented with using EA methods for the preparation of AGs. (**A**) 1,2-Epoxybutane, (**B**) propylene oxide, (**C**) *cis*-2,3-epoxybutane, (**D**) glycidol (X = OH) and epihalohydrines (X = F, Cl, Br), (**E**) trimethylene oxide, and (**F**) 3,3-dimethyloxetane. Both in the previous paper [37] and here, the image has been produced by the author using ChemDraw Ultra 7.0 software.

**Figure 4 ijms-26-10696-f004:**
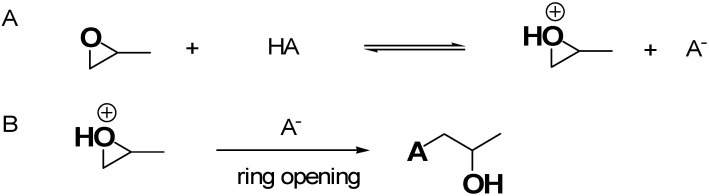
Mechanism of the EA approach using PO as the epoxide compound. (**A**) Protonation of PO by HA; (**B**) ring opening reaction via nucleophilic attack. Both in the previous paper [37] and here, the image has been produced by the author using ChemDraw Ultra 7.0 software.

**Figure 5 ijms-26-10696-f005:**
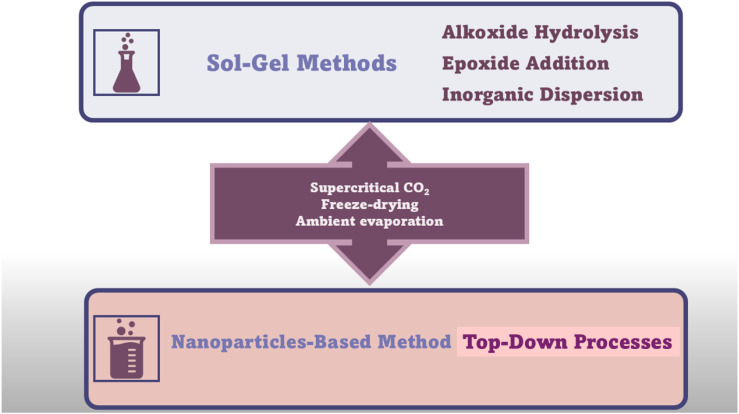
Synthetic routes to AGs are different from chalcogenide AGs (CAGs, MCAGs, and MMCAGs) and metal AGs (NMAGs and MAGs). The image is a production of the author.

**Figure 6 ijms-26-10696-f006:**
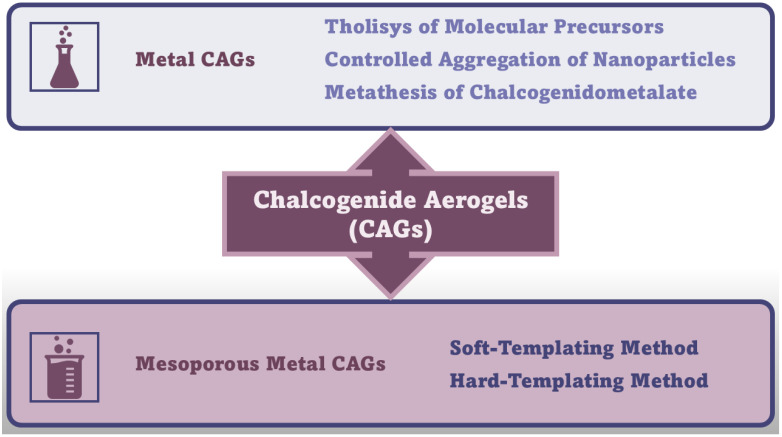
Synthetic routes to CAGs. The image has been created by the author.

**Figure 7 ijms-26-10696-f007:**
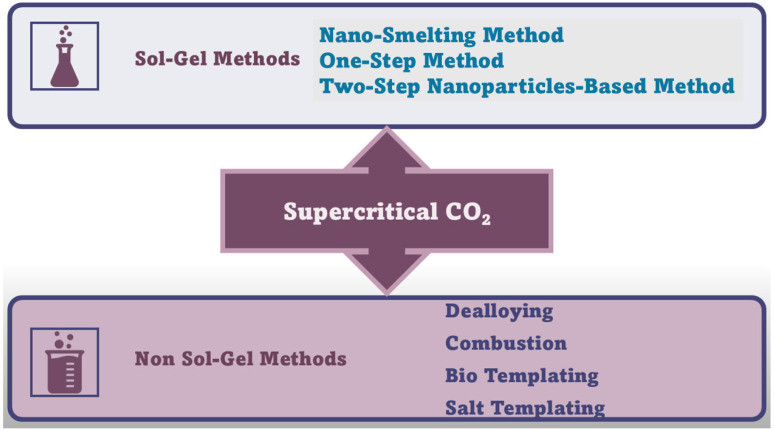
Synthetic routes to NMAGs and MAGs. The image has been created exclusively by the author.

**Figure 8 ijms-26-10696-f008:**
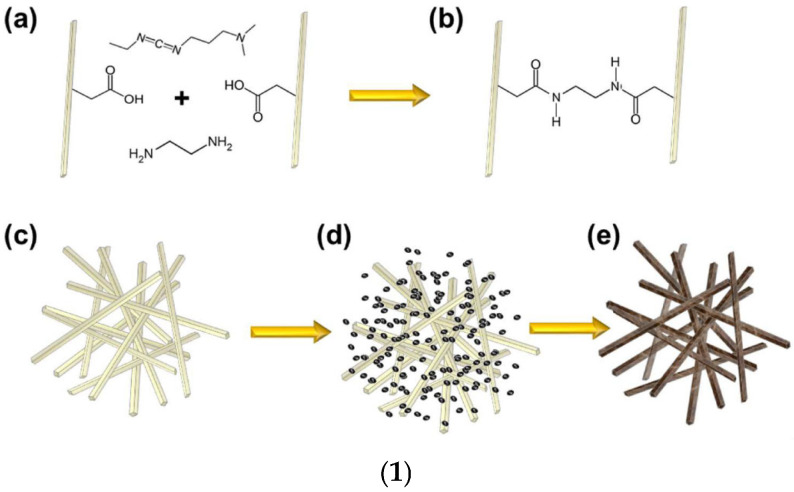
(**1**) AG synthesis scheme: (**a**) crosslinking carboxymethyl cellulose nanofibers (CNF) with EDC and ethylenediamine as a linker molecule; (**b**,**c**) crosslinked CNFs; (**d**) CNF hydrogel equilibrated with Pd salt solution; (**e**) CNF bio-templated Pd AGs after reduction with NaBH_4_, rinsing, solvent exchange with ethanol (EtOH), and supercritical CO_2_ drying. (**2**) AGs synthesis process photos: (**a**) cross linked CNF-based hydrogels with EDC and ethylenediamine as a linker molecule; CNF hydrogels equilibrated with Pd salt solutions of 1, 10, 50, 100, 500, and 1000 mM for (**b**) Pd (NH_3_)_4_Cl_2_, and (**c**) Na_2_PdCl_4_; (**d**) CNF bio-templated Pd AGs after reduction with NaBH_4_; (**e**) CNF-Pd composite AGs after rinsing, solvent exchange with EtOH and supercritical CO_2_ drying. The image was previously reported in my first-part review recently published [37], as reproduced by a MDPI open access article [327] and is licensed by the respective authors in accordance with the Creative Commons Attribution (CC-BY) license (Deed—Attribution 4.0 International—Creative Commons), which allows for unlimited distribution and reuse as long as appropriate credit is given to the original source and any changes made compared to the original are indicated.

**Table 1 ijms-26-10696-t001:** Subclassification of inorganic AGs.

Aerogels (AGs)	Sub Classes	Description/Subtypes	Refs
IAGs	SAGs	N.R.	[9,23]
	CAGs	SCAGs	[9,24]
	OAGs	SnO_2_, W_2_O_5_, ZrO_2_, TiO_2_, Al_2_O_3_, MgO, Co_3_O_4_, NiO	[25,26,27,28]
	CBAGs	SC/AGs, CDC/AGs, MC/AGs, CHC/AGs, HAN/AGs, TIC/AGs, TUC/AGs	[29,30,31]
	NAGs	CN/AGs, AN/AGs, BN/AGs, SIN/AGs	[32,33,34,35]
	MAGs/NMAGs	INs, GNs, SNs	[36]

IAGs = inorganic AGs; SAGs = silica-based AGs; CAGs = chalcogenide AGs; OAGs = oxide-based AGs; CBAGs = carbide-based AGs; NAGs = nitrite-based AGs; MAGs = metal-based AGs; NMAGs = noble metal-based AGs; SCAGs = sulfide chalcogenide AGs; SC = silicon carbide; CDC = carbon derived carbides; MC = molybdenum carbides; CHC = chromium carbide, HAN = hahnium carbide, TIC = titanium carbide, TUC = tungsten carbide; CN = carbon nitrides; ALN = aluminum nitrides; BN = boron nitrides; SIN = silicon nitride; INs = iron nanofoams; GNs = gold nanofoams; SNs = silver nanofoams; Ns Sn = Tin, W = tungsten; Zr = zirconium; Ti = titanium; Al = aluminum; Mg = magnesium, Co = cobalt; Ni = nickel; Mo = molybdenum; B = boron; Fe = iron, Au = gold; Ag = silver; N.R. = not reported.

**Table 2 ijms-26-10696-t002:** Materials used to prepare AGs via the EA method (reproduced from our recent paper [50]).

Formula	Chemical Name	Applications	Refs
TiO_2_	Titanium oxide	N.R.	[52]
V_2_O_5_	Vanadium oxide	N.R.	[53]
Co_3_O_4_	Cobalt oxide	Supercapacitors	[54]
UO_3_	Uranium trioxide	N.R.	[55]
Gd_2_O_3_	Gadolinium oxide	N.R.	[56]
La_2_O_3_	Lanthanide oxides	N.R.	[57]
ZnO_2_/TiO_2_, SiO_2_/TiO_2_	Zinc/titanium oxide, silica/titanium oxide	N.R.	[58]
Ta_2_O_5_	Tantalum oxide	N.R.	[59]
Mn_3_O_4_	Manganese oxide	N.R.	[60]
Y_2_O_3_	Yttrium oxide	N.R.	[61]
Eu-doped Y_2_O_3_	Europium-doped yttrium oxide	Luminescence	[62]
Eu-doped ThO_2_	Europium-doped oxide	N.R.	[63]
NiO/Al_2_O_3_	Nickel oxide/alumina oxides	Hydrogen production	[64]
ZnFe_2_O_4_	Zinc-ferrite oxides	N.R.	[65,66]
VFe_2_O_x_	Vanadium-ferrite oxides	Electrochemical charge storage	[67]
La_0.85_Sr_0.15_MnO_3_	Lanthanum-strontium-manganese oxides	Electronic conductivity	[68]
MnFe_2_O_4_	Manganese-ferrite-oxide	Magnetism	[69]
NiFe_2_O_4_	Nikel-ferrite oxide	Magnetic sector	[70]
Zn_5_(OH)_8_Cl_2_·H_2_O	Simonkolleite	Photoluminescence	[71]

N.R. = Not reported. La = lanthanum; Sr = strontium; Mg = manganese.

**Table 3 ijms-26-10696-t003:** Comparison between the physicochemical properties, synthesis, structures, and applications of recently developed MMCAGs and MCAGs.

Characteristics	MMCAGs	MCAGs
SF	⬆ Polarizable surface, ⬇ Lewis’s basicity	⬆ PS, ⬇ Lewis’s basicity
Synthesis	Soft-templating, hard-templating	Thiolytic hydrolysis, NPC, metathesis
Crystallinity	Crystalline to amorphous *	Nanocrystalline or amorphous
Porosity	Mesopores, narrow PSD, ⬆ SSA (516 m^2^/g)	Broad PSD, ⬆ SSA (755 m^2^/g)
AEE	Catalyst (HER/OER), PTC H_2_ production	Catalyst (HER, OER, CP), WR, AR
**References**	[111,112,113,114,115,116]	[108,109,117,118,119,120,121]

SF = surface functionality; SSA = specific surface area; AEE = application for energy and environment; MMCs = mesoporous metal chalcogenides; CBAs = chalcogenide-based aerogels; * depending on synthesis; PSD = pore size distribution; NPC = NPs condensation; HER = hydrogen evolution reaction; OER = oxygen evolution reaction; PTC = photocatalytic; CP = chemical processes; ⬆ = highly; ⬇ = soft; WR = water remediation; AR = air remediation; PS = polarizable surface.

**Table 4 ijms-26-10696-t004:** Removal of various heavy metal cations from aqueous solutions by KCMS (0.01 g). Contact time = 1 h; Solution volume =10.0 mL; Volume/mass ratio = 1000 mg/L. [149].

M*^n^*^+^	*C*_i_ (ppm)	*C*_f_ (ppb)	M*^n^*^+^ Removal (%)	*K*_d_ (mL g^−1^)
Cu^2+^	10 × 10^3^	872.0	91.30	1.05 × 10^4^
Hg^2+^	10 × 10^3^	236.8	97.63	4.12 × 10^4^
Ag^+^	10 × 10^3^	3.25	99.97	3.08 × 10^6^
Pb^2+^	10 × 10^3^	0.10	∼100.0	∼10^8^
Cd^2+^	10 × 10^3^	9.91 × 10^3^	0.86	8.65
Ni^2+^	10 × 10^3^	9.12 × 10^3^	8.8	96.49
Zn^2+^	10 × 10^3^	10 × 10^3^	0.0	0.0

**Table 5 ijms-26-10696-t005:** Comparison of adsorption capacities of KCMS for heavy metals with those of other high-performing sorbents [149].

Cations	Adsorbents	*q*_m_ (mg g^−1^)	Refs
Ag^+^	**KCMS**	**1377**	[149]
	Amorphous MoO_x_	2605	[150]
	LDH-Mo_3_S_13_	1074	[151]
	LDH-Sn_2_S_6_	978	[152]
	Ni/Fe/Ti-MoS_4_-LDH	856	[153]
	Mn-LDH-MoS_4_	564	[154]
	MoS_4_-ppy	480 (pH~5) 725 (pH∼1)	[155]
	Mo_3_S_13_-ppy	408	[156]
	MoS_4_-LDH	450	[157]
	KMS-2	408	[158]
	Fe-MoS_4_	565	[159]
Pb^2+^	**KCMS**	**1146**	[149]
	Lignosulfonate-modified graphene hydrogel	1210	[160]
	LDH-Sn_2_S_6_	579	[152]
	MoS_4_-LDH	290	[157]
	Mn-MoS_4_	357	[154]
	Fe-MoS_4_	345	[159]
	EDTA-LDH	180	[161]
	CTS/PAM gel	138	[162]
	Mg_2_Al-LS-LDH	123	[163]
	Cellulose-based CAGs	240	[164]
	Biomass-based hydrogel	422.7	[165]
Hg^2+^	**KCMS**	**460**	[149]
	LDH-Sn_2_S_6_	666	[152]
	MoS_4_-LDH	500	[157]
	Mn-MoS_4_	594	[154]
	Fe-MoS_4_	582	[159]
	KMS-2	297	[158]
	MoS_4_-ppy	210	[155]
	KMS-1	377	[166]
	Thio-functionalized magnetic graphene oxide	289	[167]

In the Table rows, bold was used to indicate adsorbents developed by Nie et al. [149].

**Table 6 ijms-26-10696-t006:** Results concerning the KCMS absorption performance of seven cations from potable Tap Water and Mississippi River Water (MRW), appropriately dissolved at 10 ppm concentration each (70 ppm total). *C*_i_ = initial (pre-sorption) concentration, and *C*_f_ = final (post-adsorption) concentration * [149].

Tap Water						MRW			
MIs	*C*_i_ (ppm)	*C*_f_ (ppm)	Removal **	*K*_d_ (mL/g)	*q*_m_ (mg/g)	*C*_f_ (ppm)	Removal **	*K*_d_ (mL/g)	*q*_m_ (mg/g)
Cu^2+^	10.0	3.6184	63.81	1782.4	6.381	3.1254	68.74	2.22 × 10^3^	6.874
Hg^2+^	10.0	0.2132	97.86	4.60 × 10^4^	9.786	0.1749	98.25	5.68 × 10^4^	9.825
Ag^+^	10.0	0.0336	99.66	2.96 × 10^5^	9.966	0.0524	99.48	2.08 × 10^5^	9.948
Pb^2+^	10.0	0.0686	99.31	1.51 × 10^5^	9.931	0.0279	99.72	3.4 × 10^7^	9.972
Cd^2+^	10.0	8.0149	19.85	258.65	1.985	8.1857	18.14	2.30 × 10^2^	1.814
Ni^2+^	10.0	9.5000	4.64	48.63	0.463	9.3766	6.23	6.63 × 10^1^	0.623
Zn^2+^	10.0	10	0.0	0.0	0.0	10.0	0.0	0.0	0.0

* Contact time: 24 h; V = 10.0 mL; m (mass of KCMS) = 0.01 g; V/m ratio = 10/0.01 = 1000 mL/g; MIs = mixed ions; ** = %.

**Table 7 ijms-26-10696-t007:** Parameters of PSO kinetics for the adsorption of RhB, MB, and MV on SbS [168].

Dyes	*C*_o_ (mg/100 mL)	*Q*_e,exp_ (mg/g)	*K*_2_ g mg^−1^ min^−1^	*Q*_e,cal_ (mg/g)	*R* ^2^
RhB	100	416.33	0.0028	418.41	0.9997
MB	100	260.00	0.0072	263.15	0.9992
MV	100	280.89	0.0066	280.11	0.9983

**Table 8 ijms-26-10696-t008:** Isotherm parameters of Langmuir and Freundlich models for adsorption of RhB, MB, and MV on SbS [168].

Adsorbent	Dyes	Langmuir				Freundlich		
		AC *q*_m_ (mg/g)	*K*_L_ (L/mg)	*R* ^2^	*R* _L_	*n*	*K*_f_ (L/mg)	*R* ^2^
SbS	RhB	442.47	0.02181	0.9840	0.0353	3.09	11.03	0.7223
	MB	303.95	0.08756	0.9956	0.0090	8.74	12.83	0.7174
	MV	210.52	0.06157	0.9751	0.0128	9.11	12.96	0.7149

**Table 9 ijms-26-10696-t009:** Comparison of Langmuir adsorption capacities of SbS towards RhB, MB, and MV with other adsorbents [168].

Dyes	Adsorbent	*q*_max_ (mg/g)	Ref
RhB	Molybdenum sulfide (MoS_2_)	291	[169]
	PDA-modified graphene hydrogel	207.06	[170]
	Activated carbon	88.0	[171]
	CK-30 (COOH functionalized KIT-5)	270	[172]
	Hydrothermal graphene hydrogel	148.41	[170]
	Unmodified biomass	25.2	[173]
	A-TRB	212.77	[174]
	Fe–Ben	98.62	[175]
	NMIL-100 (Fe)	76.69	[176]
	MIL-68 (Al)	1111.11	[177]
	**Antimony sulfide (SbS)**	**442.47**	[168]
MB	MoS_2_	208	[169]
	Carbon monolith	127.06	[178]
	Cobalt ferrite/alginate	33.6	[179]
	CMT-*g*-PAM/silica nanocomposite	43.86	[180]
	Magnetic graphene oxide	270.94	[181]
	Swede rape straw C2-symmetric benzene-based	246.4	[182]
	Hydrogels with unique layered structures	54.4	[183]
	Superabsorbent hydrogel supported on modified polysaccharide	48	[184]
	Poly(AA-*co*-AMPS)MMT	215	[185]
	MIL-68 (Al)	1666.67	[177]
	MIL-68 (Al)	666.67	[186]
	**SbS**	**303.95**	[168]
MV	Baker’s yeast modified by nano Fe_3_O_4_	60.8	[187]
	Sunflower seed hull	92.6	[188]
	Magnetic baker’s yeast biomass	60.80	[187]
	Granular activated carbon	95	[189]
	Bagasse fly ash	26.2	[190]
	Peanut straw char	101	[190]
	Halloysite nanotubes	113.6	[191]
	Cross-linked amphoteric starch	333.3	[192]
	Sugar cane dust	50.4	[193]
	**SbS**	**210.52**	[168]

In the Table rows, bold was used to indicate adsorbents developed by Raju et al. [168].

**Table 10 ijms-26-10696-t010:** Ion-exchange capacities of MCIEs containing alkali metal ions in removing (RI) [194].

Material	*T*	Target Ion	*K*_d_ (mL/g)	*q*_m_ (mg/g)	*T* (min)	pH Range	Ref.
K_2*x*_Mn_x_Sn_3−x_S_6_ (*x* = 0.5–0.95) **(KMS-1)**	RT	Sr^2+^	≥10^5^	77	——	1.0–14	[195]
		Cs^+^	≥10^4^	226	5	0.8–12	[196]
		UO_2_^2+^	≥10^5^	380	120	2.5–10	[197]
K_2x_Mg_x_Sn_3−x_S_6_ (*x* = 0.5–1) **(KMS-2)**	RT	Cs^+^	≥10^3^	531	——	3.0–10	[198]
		Sr^2+^	≥10^4^	87			
KInSn_2_S_6_ **(KMS-5)**	RT	Eu^3+^	≥5.91 × 10^4^	87	10	1.0–5	[199]
		^241^Am	——	——			
		Th^4+^	1.5 × 10^6^	90.9	10	——	[200]
KInSnS_4_ **(InSnS-1)**	RT	Cs^+^	>10^4^	316.0	5	0.02–11.14 2 mol/L HNO_3_	[201]
K_1.87_ZnSn_1.68_S_5.30_ **(KZTS)**	298 K	Sr^2+^	1.28 × 10^6^	19.3	10	3.0–11	[202]
Na_5_Zn_3.5_Sn_3.5_S_13_·6H_2_O **(NaZTS)**	298 K	Sr^2+^	2.67 × 10^5^	32.3	5	3.0–12	[203]
Na_5_Zn_3.5_Sn_3.5_S_13_·6H_2_O **(ZnSnS-1)**	343 K	Sr^2+^	1.6 × 10^4^	124.2	1440	2.5–13	[204]
K_2x_Sn_4−x_S_8−x_ (*x* = 0.65–1) **(KTS-3)**	RT	Cs^+^	5.5 × 10^4^	280	5	2.0–12	[205]
		Sr^2+^	3.9 × 10^5^	102		2.0–12	
		UO_2_^2+^	2.7 × 10^4^	287		4.0–10	
K_2_Sn_2_S_5_ **(KTS-2)**	343 K	Yb^3+^	>10^5^	232.7	10	——	[206]
Na_2_Sn_3_S_7_ **(NaTS)**	298 K	Sr^2+^	3.43 × 10^7^	80	5	3.0–13	[207]
Na_1.94_Sn_2.87_S_7_ **(NaTS-2)**	298 K	Sr^2+^	2 × 10^6^	88.9	60	3–11	[208]
Na_1.60_Mg_0.33_Sn_3.15_S_7_ **(NMTS)**	298 K	Sr^2+^	1.1 × 10^6^	52.6	3	4–11	[209]
K_1.93_Ti_0.22_Sn_3_S_6.43_ **(KTSS)**	RT	Cs^+^	>10^4^	450.12	1	3–12	[210]
K_1.61_Fe_0.04_Sb_0.03_Sn_3.1_S_7_ **(PIATS)**	RT	Cs^+^	1.96 × 10^4^	401.23	3	4–12	[211]
K_1.29_Sb_0.15_Sn_3_S_6.87_ **(KATS-2)**	RT	Cs^+^	1.59 × 10^5^	358	5	1–12	[212]
K_1.83_Al_0.48_Sn_3_S_7.64_ **(KAlSnS-3)**	298 K	Cs^+^	2.09 × 10^5^	259.31	15	1–13	[213]
KInSnS_4_	RT	NH_4_^+^, Rb^+^, Cs^+^,Tl^+^, Sr^2+^, Ca^2+^, Ln^3+^					[214]
NaInSnS_4_	RT	NH_4_^+^, Rb^+^, Cs^+^,Tl^+^, Sr^2+^, Ca^2+^, Ce^3+^					
K_2_CdSn_2_S_6_	RT	Rb^+^, Cs^+^,Tl^+^, Sr^2+^, Ce^3+^					[215]
K_3.2_Nb_1.6_S_4.8_ **(KNbS)**	303 K	Sr^2+^	——	80	25	4–10	[216]
		Co^2+^	——	55	35	3.5–12	
K_6_Cd_4_Sn_3_Se_13_	——	Li^+^, Na^+^, Rb^+^					[217]
K_3_Rb_3_Zn_4_Sn_3_Se_13_	323 K	Cs^+^					[218]

**Table 11 ijms-26-10696-t011:** Ion-exchange capabilities of MCIEs containing protonated organic amine cations [194].

Compounds	*T*	Target Ion	*K*_d_ (mL/g)	*q*_m_ (mg/g)/RR (%)	*T* (min)	pH
FJSM-SnS	338 K	Cs^+^	2.36 × 10^3^	408.91	5	0.7–12.7
		Sr^2+^	8.89 × 10^4^	65.19 ^RR^	5	
	RT	UO_2_^2+^	2.64 × 10^4^	338.43	1200	2.1–11
	338 K	Eu^3+^	>10^4^	139.82	<5	1.9–8.5
		Tb^3+^		147.05	<5	
		Nd^3+^		126.70	——	
	298 K	Ba^2+^	7.49 × 10^4^	289.0	<5	3.3–10.8
		Ni^2+^	8.92 × 10^4^	83.27 ^RR^		2.3–10.8
		Co^2+^	3.75 × 10^5^	51.98 ^RR^		
FJSM-SnS-2	RT	Cs^+^	1.81 × 10^3^	266.54	60	2.5–11.9
		Sr^2+^	4.47 × 10^3^	59.41 ^RR^	60	
		Eu^3+^	>10^5^	58.39 ^RR^	5	
FJSM-SnS-3	RT	Cs^+^	2.62 × 10^3^	109.68	60	
		Sr^2+^	3.59 × 10^3^	57.81 ^RR^	60	
		Eu^3+^	>10^5^	61.52 ^RR^	5	
FJSM-SnS-4	RT	Cs^+^	5.83 × 10^3^	388.94	30	0.0–11.1
		Sr^2+^	9.38 × 10^5^	141.22	2	
InS-0	——	Sr^2+^	——	∼59.6 ^RR^	60	——
		Cs^+^	——	∼41.2 ^RR^	——	——
InS-1	RT	Sr^2+^	1.57 × 10^5^	105.35	1440	3.0–12
InS-2	RT	Sr^2+^	>10^4^	143	480	3.0–14
		Ba^2+^	2.3 × 10^5^	211.73	20	3–13
		Ni^2+^	2.0 × 10^5^	103.57	20	3–11
		Co^2+^	1.6 × 10^5^	111.78	10	3–11
(NH_4_)_4_In_12_Se_20_	RT	Cs^+^Sr^2+^	——	——	——	——
FJSM-SbS	353K	Cs^+^	5.62 × 10^3^	143.47	2	3.4–11.4
[(Me)_2_NH_2_]_2_[Sb_2_GeS_6_]	RT	Cs^+^	——	∼93 ^RR^	——	——
		Rb^+^	——	∼85 ^RR^	——	——
[CH_3_NH_3_]_20_Ge_10_Sb_28_S_72_·7H_2_O	338 K	Cs^+^	5.46 × 10^3^	230.91	2	2.8–11
[(CH_3_CH_2_CH_2_)_2_NH_2_]_5_In_5_Sb_6_S_19_·1.45H_2_O	348 K	Cs^+^	——	228.61	——	4.3–10
[NH_3_CH_3_]_4_[In_4_SbS_9_SH]	343 K	Rb^+^	——	∼82.5 ^RR^	——	——
[(CH_3_)_2_NH_2_]_2_[Ga_2_Sb_2_S_7_]·H_2_O	348 K	Cs^+^	——	100 ^RR^	——	1.7–11.8
FJSM-GAS-1	RT	Cs^+^	——	164	——	——
		Sr^2+^	——	80 ^RR^	——	——
		UO_2_^2+^	2.47 × 10^4^	196	5	2.9–10
		Eu^3+^	6.39 × 10^5^	127.7	<2	2.5–8.2
FJSM-GAS-2	RT	UO_2_^2+^	5.12 × 10^4^	144	15	2.9–10
		Eu^3+^	6.00 × 10^5^	115.8	<2	2.5–8.2
SCU-36	RT	Cs^+^	——	∼70 ^RR^	——	——
InSnOS	RT	Cs^+^	10^5^	537.7	5	4.0–10
[(Me)_2_NH_2_]_0.75_ [Ag_1.25_SnSe_3_]	RT	Cs^+^	——	93 ^RR^	——	——
		Rb^+^	——	87 ^RR^	——	——
CuGeSe-1	343 K	Cs^+^	3.5 × 10^3^	225.3	——	1.0–12
AgSnSe-1	RT	Cs^+^	3.17 × 10^4^	174.4	——	1.0–12
[CH_3_CH_2_NH_3_]_22_Zn_16_Sn_12_Se_51_(H_2_O)_4_·16H_2_O	RT	Sr^2+^	>10^4^	104.17	20	——
(H_2_en)_2_Cu_8_Sn_3_S_12_	333 K	Cs^+^	——	∼35 ^RR^	——	——
	333 K	Rb^+^	——	∼36 ^RR^	——	——
(dap)_2_(Hdap)_4_Cu_8_Ge_3_S_18_	318 K	Cs^+^	——	90 ^RR^	——	——
OCF-45-MnInS	RT	Cs^+^	——	90 ^RR^	——	——
	333 K	Cs^+^	——	100 ^RR^	——	——
CdSnSe-1	RT	Cs^+^	1.42 × 10^4^	371.4	1440	4–13
		Sr^2+^	1.50 × 10^4^	128.4	1440	3–13
[(Me)_2_NH_2_] [BiGeS_4_]	RT	Rb^+^	——	25 ^RR^	——	——
UCR-28	RT	Sr^2+^	——	90	90–120	——

RR = removal rate; FJSM-SnS = (Me_2_NH_2_)_1.33_(Me_3_NH)_0.66_Sn_3_S_7_·1.25H_2_O; FJSM-SnS-2 = [CH_3_NH_3_] [Bmmim]Sn_3_S_7_·0.5H_2_O; FJSM-SnS-3 = [CH_3_NH_3_]_0.75_ [Bmmim]_1.25_Sn_3_S_7_·H_2_O; FJSM-SnS-4 = [(EtNH_3_)_1.68_(Et_2_NH_2_)_0.32_]Sn_3_S_7_·0.68H_2_O; InS-0 = [In_10.5_S_14.5_]·[(H_2_NCH_2_CH_2_-NHCH_2_)_2_]_2.5_; InS-1 = [CH_3_CH_2_NH_3_]_6_In_6_S_12_; InS-2 = [CH_3_CH_2_NH_3_]_6_In_8_S_15_; FJSM-SbS = [MeNH_3_]_3_Sb_9_S_15_; FJSM-GAS-1 = [Me_2_NH_2_]_2_ [Ga_2_Sb_2_S_7_]·H_2_O; FJSM-GAS-2 = [Et_2_NH_2_]_2_ [Ga_2_Sb_2_S_7_]·H_2_O; SCU-36 = (Hdmp)_4_·In_3_SnS_8.5_; InSnOS = (Heta)_9.5_(H_3_O)_2.5_ [In_8_Sn_12_O_10_S_32_]·22H_2_O; CuGeSe-1 = [NH_3_CH_3_]_0.75_Cu_1.25_GeSe_3_; AgSnSe-1 = [NH_3_CH_3_]_0.5_ [NH_2_(CH_3_)_2_]_0.25_Ag_1.25_SnSe_3_; OCF-45-MnInS = [(In_43_Mn_11_S_87_)]·12(H^+^-DBU)·11(H^+^-PR)·5.1H_2_O; CdSnSe-1 = [CH_3_NH_3_]_3_ [NH_4_]_3_Cd_4_Sn_3_Se_13_·3H_2_O; UCR-28 = [Zn(C_6_N_4_H_18_)(H_2_O)] [Ge_3_S_6_Zn(H_2_O)S_3_Zn(H_2_O)].

**Table 12 ijms-26-10696-t012:** Ion-exchange capacities of MCIEs prepared by the stepwise activation/intercalation method [194].

Material	*T*	Target Ion	*K*_d_ (mL/g)	*q*_m_ (mg/g) or RR (%)	*T* (min)	pH Range	Ref.
K@RWY	RT	Cs^+^	≥10^5^	310	5	2.9–11.8	[219]
K@GaSnS-1	RT	UO_2_^2+^	1.03 × 10^4^	147.6	15	2.75–10.87	[220]
KIAS	RT	Cs^+^	>10^4^	309.6	1	1–13	[221]
SbS-1	RT	Cs^+^	N.R.	70.96 ^RR^	1440	N.R.	[222]
		Sr^2+^		49.48 ^RR^	>2880		
SbS-1K	RT	Cs^+^	1.06 × 10^5^	318.77	40	0–12	[222]
		Sr^2+^	1.95 × 10^5^	61.12 ^RR^	40	4–11	
K-MPS-1	RT	Cs^+^	>10^4^	337.5	15	2–12	[223]
N-MPS	RT	UO_2_^2+^	2.23 × 10^4^	854.36	360	2.8–12.2	[224]

N.R. = Not reported; RR = removal rate; RT = room temperature; KIAS = K_2_In_2_Sb_2_S_7_·2.2H_2_O; SbS-1 = (NH_4_)_2_Sb_4_S_7_; SbS-1K = K_2_Sb_4_S_7_·2H_2_O; K-MPS-1 = K_0.48_Mn_0.76_PS_3_·H_2_O; N-MPS = (NH_4_)_0.48_Mn_0.76_PS_3_⋅H_2_O.

**Table 13 ijms-26-10696-t013:** Synthetic methods, morphologies, and compositions of various NMAGs [288].

MT	Synthetic Methods	Morphologies	Compositions	Refs
TSM	C_2_H_5_OH or H_2_O_2_	NWs	Au, Pt, Ag, PtAg, AuAg	[270]
	348 K, GRR	HNWs	Ni–Pd_x_Pt_y_	[294]
	Cu-UPD	CSNWs	Pd_x_Au–Pt	[329]
	348 K, GRR	HNWs	PdNi	[295]
	Salting out	NWs	Au, Ag, Pd, Pt	[314]
	Freeze–thaw		Au, Pd, Rh, AuAg, AuPd, AuPt, AuRh	[330]
	Excessive-reductant-directed		Au, Ag, Pd, Pt, Ru, Rh, Os, Ir	[315]
	NaCl		Au*^m^*^−*n*^Pd	[331]
			Au	[332]
	Ligand-exchange		Au	[333]
	NaBH_4_		Au, Pd, AuPd	[326]
	C(NO_2_)_4_	HNWs	Ag	[291]
			AuAg alloy	[287]
	348 K	NWs	Au, Ag, Pt, Pd	[290]
	Phase boundary gelation	NMs	Au	[334]
	C(NO_2_)_4_	NWs	Au/Ag/Pd	[304]
	N_2_H_4_·H_2_O	NCs	Pt	[273]
	NaCl	NSs	Au/Ag, Pd/Ag, Pt/Ag	[94]
	AA, 35 °C	DCSs	Au@Pt_3_Pd	[335]
	NaBH_4_, 60 °C, GRR	NWs	AD-Pt@AuCu	[336]
			Au_2_Cu@Pd	[322]
	Ca^2+^		Pd	[313]
	DA		Au	[337]
	NaBH_4_, 25 °C, GRR	NTs	PtAg	[338]
OSM	NaBH_4_, 25 °C	NWs	Pd_CD_	[296]
	DMAB, NaHPO_2_·H_2_O, 25 °C		Au, Pd, Pt	[339]
	NH_4_F		AuPt, AuRh	[275]
	Stirring		Au, Ag, Pd, Rh, Au–Ir, Au–Pd–Pt	[324]
	NaBH_4_, 25 °C		PtNi	[323]
			Pt_x_Pd*_y_*	[274]
	CHOCOOH, 45 °C	NWs	Pd	[340]
	NaBH_4_, 60 °C	NVNWs	Ir_x_Cu	[320]
		NWs	IL/PdCu	[319]
			AgPd-Pt_dilute_	[341]
			AuPt	[298]
	NaBH_4_, DA, 60 °C	NWs	AuPt-PDA	[280]
	CO, 50 °C		Pd	[342]
	NaBH_4_, 60 °C		PdCu	[297]
	NaH_2_PO_2_, 60 °C	CSINs	PdPb@Pd	[343]
	NaBH_4_, 60 °C	NWs	PtRuCu	[321]

MT = Method type; TSM = two-step method; OSM = one-step method; CSINs = core–shell intermetallic nanowires; NWs = nanowires; HNWs = hollow nanowires; CSNWs = core–shell nanowires; NMs = nanomeshes; NCs = nanocubes; NSs = Nanoshells, DCSs = dendritic core–shells; NTs = nanotubulars; NVNWs = nanovoid nanowires.

**Table 14 ijms-26-10696-t014:** Main sectors of application of AGs with motivations and possible uses and/or deriving advantages.

Sector	Possible Application	Motivations	Examples/Advantages	Refs
Biomedicine	Controlled DDSs	Tuneable PS, ⬆ SSA, CRR of PP	SAGs for targeted CT *	[361]
Energy	Batteries, supercapacitors	⬆ SSA	⬆ Charge storage capacities **	[81,362]
AE	Insulation	IP and resistance to ET	Spacecraft/satellites protection **	[363]
EE	FS for AP and WP	⬆ Trap particulates/contaminants	HPFS for EP **	[364].

* Industrial applications have already hit the market concerning drug delivery, biocatalysis, and biosensing; ⬆ = high, higher; ** marketed, EE = Environmental engineering; AE = aerospace engineering; DDDs = drug delivery systems; FS = filtration systems; AP = air purification; WP = water purification; SSA = specific surface area; PP = pharmaceutical principle; PS = pore sizes; CRR = controlled release rate; HPFS = high-performance filtration solution; IP = insulation properties; ET = extreme temperatures; CT = cancer therapy, EP = environmental pollution.

## Data Availability

No new data were created or analyzed in this study. Data sharing is not applicable to this article.

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
