# Peer review of "Aerogels Part 2. A Focus on the Less Patented and Marketed Airy Inorganic Networks Despite the Plethora of Possible Advanced Applications"

_ijms, 2025, doi:10.3390/ijms262110696_

Round 1
Reviewer 1 Report (New Reviewer)
Comments and Suggestions for Authors
The manuscript offers a comprehensive and well-structured review of inorganic aerogels, with particular emphasis on less explored categories such as chalcogenide-based and metal-based aerogels. The topic is highly relevant and timely, and the author has effectively compiled and synthesised a broad and pertinent body of literature. Including detailed synthetic strategies, property comparisons, and potential applications makes this work a valuable contribution to the field.
Nonetheless, I would like to suggest two ways to further enhance the quality and clarity of the manuscript:
- While the technical content is excellent, the manuscript contains numerous grammatical errors and awkward sentence structures that sometimes impede readability. For example, the sentence “AGs have been applied mainly as thermal insulators” (line 41) could be better phrased as “AGs have primarily been used as thermal insulators”.
- Phrases like “the due background about IAGs” (line 17) sound unnatural in scientific English and could be replaced with “a brief background on IAGs”.
- The manuscript references numerous supplementary figures (S1–S11), many of which are listed sequentially without deeper discussion or interpretation. This approach may overwhelm readers and detract from the narrative coherence. Consider moving the most informative or representative figures into the main text to support key discussions.
- Provide more analytical commentary alongside each figure, highlighting major trends, insights, or implications rather than limiting the description to visual features alone.
A thorough language revision by a native English speaker or a professional editor experienced in scientific writing is highly recommended to improve clarity and fluency.
Author Response
The manuscript offers a comprehensive and well-structured review of inorganic aerogels, with particular emphasis on less explored categories such as chalcogenide-based and metal-based aerogels. The topic is highly relevant and timely, and the author has effectively compiled and synthesised a broad and pertinent body of literature. Including detailed synthetic strategies, property comparisons, and potential applications makes this work a valuable contribution to the field.
I thank the Reviewer for her/his positive comments and for having appreciate my review work.
Nonetheless, I would like to suggest two ways to further enhance the quality and clarity of the manuscript:
- While the technical content is excellent, the manuscript contains numerous grammatical errors and awkward sentence structures that sometimes impede readability. For example, the sentence “AGs have been applied mainly as thermal insulators” (line 41) could be better phrased as “AGs have primarily been used as thermal insulators”.
I thank the Reviewer for this valuable suggestion. The sentence has been modified according to the Reviewer’s comment. Please, see at lines 44-45.
- Phrases like “the due background about IAGs” (line 17) sound unnatural in scientific English and could be replaced with “a brief background on IAGs”.
I thank the Reviewer for this valuable suggestion. The sentence has been modified according to the Reviewer’s comment. Please, see at lines 17-18 and 90-91.
- The manuscript references numerous supplementary figures (S1–S11), many of which are listed sequentially without deeper discussion or interpretation. This approach may overwhelm readers and detract from the narrative coherence. Consider moving the most informative or representative figures into the main text to support key discussions.
The Reviewer suggestion is valuable and rational. Anyway, my choice to insert Figure S1-S11 as well in a Supplementary Materials file derives from a similar choice made for Aerogel Part 1, already published on Gels and to maintain continuity among the two works, which are strictly connected, despite will result published on different journals. Note that the first decision was made to not further burden the file of the main text of previous review, already difficult to be open with additional very large images. Especially, when the main text already contained up to ten Figures. Therefore, I kindly ask the Reviewer to not force me to move Supplementary Figures inside the main text, and to accept my choice.
- Provide more analytical commentary alongside each figure, highlighting major trends, insights, or implications rather than limiting the description to visual features alone.
Thank you so much to the Reviewer for her/his valuable suggestion. As asked, more analytical commentary and numerical information alongside each figure have been added for all Figures and major trends, insights, or implications have been highlighted when observable.
Comments on the Quality of English Language
A thorough language revision by a native English speaker or a professional editor experienced in scientific writing is highly recommended to improve clarity and fluency.
I thank the Reviewer for her/his valuable comment, which has enabled me to improve the English language of this paper. On suggestion of the Reviewer, the manuscript has been revised by prof. Deirdre Kantz, English mother tongue teacher, my colleague, who works for University of Genoa and Pavia, to improve language.
Reviewer 2 Report (New Reviewer)
Comments and Suggestions for Authors
This is an interesting review paper focusing on chalcogenide and metallic aerogels, with a wealth of references. The most well-written section is that on chalcogenide aerogels.
However, the introductory portion (roughly the first ten pages) has major issues and will require substantial revision. In particular, the description of the drying process is largely incorrect, and the example of wood aerogels is not successful. The coverage of oxide and carbide aerogels is also incomplete.
The classification of aerogels around Figure 1 presents major conceptual issues and should be reconsidered.
The section on metallic aerogels should be expanded to include materials prepared by carbothermal reduction and to discuss in greater detail their application as energetic materials. For instance, iron aerogels are known to function as either explosives or thermites depending on their nanostructure.
I have provided extensive comments for improvement directly on the manuscript, which I am attaching as an annotated PDF file.

Author Response
This is an interesting review paper focusing on chalcogenide and metallic aerogels, with a wealth of references. The most well-written section is that on chalcogenide aerogels.
I thank the Reviewer for her/his positive comments and for having (at her/him mode) appreciate my review work, at least in one part.
However, the introductory portion (roughly the first ten pages) has major issues and will require substantial revision. In particular, the description of the drying process is largely incorrect, and the example of wood aerogels is not successful.
I thank the Reviewer for all comments and suggestions which have been inserted as comments in the PDF provided. When possible and not too contrasting with what reported in the relevant published reviews and articles, I used to construct this my review, I revised this part as well as the whole manuscript or given explanations, which have been inserted in the PDF as responses to the Reviewer’s comments.
The coverage of oxide and carbide aerogels is also incomplete.
According to the Reviewer’s comments the lists of carbide and nitride aerogels have been completed, while that of oxides was not modified, since I have reported only some examples of the high number of oxide AG developed so far. I retain that at an introductory level, it could be enough. In addition, I make kindly note to the Reviewer that oxide and carbide aerogels did not enter in the scope of this second part review, focused on the “Less Patented and Marketed Airy Inorganic Networks”, which, according to what reported in the CAS Content Collection are CAGs and MAGs, deeply discussed here. Also, from the abstract, the Reviewer can apprehend that: “In this second-part review, IAGs were handled again, but chalcogenide and metals AGs (CAGs and MAGs) were debated, since they are still too little studied, patented and marketed, despite their nonpareil properties and vast range of possible applications”. Additionally, I make kindly note to the Reviewer that concerning oxide-based AGs, they were already in deep discussed in my first part review (Ref. 37). Finally, as I have commented elsewhere, my work wants to be a review on the selected two classes of AGs and not a new Handbook, which already exists and with a nonpareil level of information, which is not in my scope.
The classification of aerogels around Figure 1 presents major conceptual issues and should be reconsidered.
Despite original classification in Figure 1 respects classifications reported in several published papers, according to the suggestions of Reviewer contained in the comments in the PDF file provided, Figure 1 has been modified.
The section on metallic aerogels should be expanded to include materials prepared by carbothermal reduction and to discuss in greater detail their application as energetic materials. For instance, iron aerogels are known to function as either explosives or thermites depending on their nanostructure.
As required, this part has been extensively expanded following the suggestions from the Reviewer contained in the provided PDF file and citing the papers indicated by the Reviewer. I thank the Reviewer for this.
I have provided extensive comments for improvement directly on the manuscript, which I am attaching as an annotated PDF file.
All manuscript, when possible and not too in contrast with the relevant reviews, article and chapters from which I took information reported in my paper, has been modified according to comments of the Reviewer contained in the PDF file, also citing the works suggested by the Reviewer. Anyway, hoping to not disturb the Reviewer, I found her/his comments, not professional and sometime close to being offensive. With such comments the Reviewer has judged my knowledge in the field too harshly and negatively, using not professional expressions not expected by a high-level scientist as I think she/he could be. I certainly don't have the knowledge of an expert in the field like the Reviewer, but all the information reported in my paper, that the Reviewer objected to, has been taken by relevant articles in the field and they were not invented by me.
Comments on the Quality of English Language
The English could be improved to more clearly express the research.
I thank the Reviewer for her/his valuable comment, which has enabled me to improve the English language of this paper. On suggestion of the Reviewer, the manuscript has been revised by prof. Deirdre Kantz, English mother tongue teacher, my colleague, who works for University of Genoa and Pavia, to improve language.

Reviewer 3 Report (New Reviewer)
Comments and Suggestions for Authors
- The paper does not sufficiently consider the use of aerogels in medicine and pharmaceuticals. Many articles have been devoted to this application, but the author has focused on catalysis, adsorption, and wastewater treatment. I would like to see a more comprehensive review.
- The following text is written in line 180: «In freeze-drying process, ice crys-181 tals form, form which the original microstructure can be rebuilt, providing AGs with high 182 porosity, very low density, notably improved permeability, and optimal cyclic water ab-183 sorption capacity (WAC).» However, this does not correspond to reality, since it is unclear how, after the formation of ice crystals during freezing and the formation of macropores, macropores turn into micropores.
- The work does not consider the effect of different solvents on the aging stage of gels.
- I don't understand what it means: "Upon this phenomenon, NPs loses their hard sphere-like behavior becoming “stickier", thus colliding/fusing together" line 290.
- There is a lack of comparison of aerogels by numerical characteristics. In most of the article, the comparison is based on SAM's photographs. This method cannot be used for qualitative or theoretical evaluation, since its results strongly depend on the location of the images taken on the material (external surface, internal).
- The comparison lacks aerogels in terms of their numerical characteristics. Most of the article relies on SEM images for comparison. This method cannot be used for qualitative or theoretical evaluation, as its results are highly dependent on the sample location on the material (external surface or internal part). It would be interesting to see comparative values ​​for specific surface area, pore diameter, porosity, and density depending on the results of the synthesis methods and parameters.
English is satisfactory
Author Response
- The paper does not sufficiently consider the use of aerogels in medicine and pharmaceuticals. Many articles have been devoted to this application, but the author has focused on catalysis, adsorption, and wastewater treatment. I would like to see a more comprehensive review.
I thank the Reviewer for this comment. Anyway, since this is a review specifically focused on CAGs, MCAGs, MMCAGs, MAGs and NMAGs, the less patented and studied AGs, the request of this Reviewer cannot be addressed. In facts, despite AGs found a plethora of applications in the biomedical sector, AGs different from those in the scope of this review are finalized, among others, for biomedical applications. They mostly include oxide AGs and especially silica oxide AGs based on chitosan and other natural materials. Here, catalysis, adsorption, and wastewater treatment, as well as energetic applications have been discussed since they are the peculiar ones for these materials.
- The following text is written in line 180: «In freeze-drying process, ice crystals form, form which the original microstructure can be rebuilt, providing AGs with high porosity, very low density, notably improved permeability, and optimal cyclic water absorption capacity (WAC)» However, this does not correspond to reality, since it is unclear how, after the formation of ice crystals during freezing and the formation of macropores, macropores turn into micropores.
I thank the Reviewer for this comment. The sentence has been corrected. Please, see lines 187-120 and 203-206.
- The work does not consider the effect of different solvents on the aging stage of gels.
The effect of different solvent on the aging stage of gels have been considered and discussed in the revised manuscript in lines 1696-179. I thank the Reviewer for this valuable suggestion.
- I don't understand what it means: "Upon this phenomenon, NPs loses their hard sphere-like behavior becoming “stickier", thus colliding/fusing together" line 290.
I thank the Reviewer for this comment. The sentence has been modified for major clarity. Lines 316-318.
- There is a lack of comparison of aerogels by numerical characteristics. In most of the article, the comparison is based on SAM's photographs. This method cannot be used for qualitative or theoretical evaluation, since its results strongly depend on the location of the images taken on the material (external surface, internal).
The requested numerical values, at least for a physical characteristic, have been inserted for all AGs, whose SEM Images have been reported.
- The comparison lacks aerogels in terms of their numerical characteristics. Most of the article relies on SEM images for comparison. This method cannot be used for qualitative or theoretical evaluation, as its results are highly dependent on the sample location on the material (external surface or internal part). It would be interesting to see comparative values ​​for specific surface area, pore diameter, porosity, and density depending on the results of the synthesis methods and parameters.
As reported in the previous point, the requested numerical values, at least for a physical characteristic, have been inserted for all AGs, whose SEM Images have been reported.
Round 2
Reviewer 2 Report (New Reviewer)
Comments and Suggestions for Authors
Sorry if you felt offended by the delivery of my review. Look at it as constructive criticism. You put all this effort on this manuscript. You do not want to include things that may be incorrect. Bad impressions are difficult -to impossible- to reverse. Just next time be more careful.
This manuscript is a resubmission of an earlier submission. The following is a list of the peer review reports and author responses from that submission.
Round 1
Reviewer 1 Report
Comments and Suggestions for Authors
I believe this work will be highly valuable for researchers in the field. However, several aspects of the manuscript could be improved, I suggest its publication after major revisions.
Comment 1
In the Abstract of this paper, it is mentioned that “Specifically, the properties, specific synthesis including the one and two-step sol-gel methods, dealloying, combustion and bio-template processes, and their possible uses were disserted.” However, in the main text, the properties and specific synthesis methods are described in a very cursory manner.
Comment 2
Throughout the paper, there are numerous tables but only two figures, which makes the article relatively difficult to comprehend and may hinder readers' understanding of the content. It is recommended to add relevant images and schematic diagrams. For example, in Section 1 (Introduction), schematic diagrams of different aerogels can be included; in Section 3 (Synthetic Methods to Achieve AGs: A Brief Overview), real images of aerogels synthesized via the sol-gel method can be added.
Comment 3
The logic of this paper is unclear. In Section 3 (Synthetic Methods to Achieve AGs: A Brief Overview), the traditional sol-gel method for preparing aerogels is introduced. However, in Sections 4.2.1 (Sol-Gel Methods to Prepare Noble Metal Aerogels (NMAGs)) and 4.2.2 (Non-Sol-Gel Methods to Prepare Metallic Aerogels), the sol-gel and non-sol-gel methods for preparing metal-based aerogels are discussed again. Similar content should be integrated into Section 3. Additionally, the introduction of noble metal aerogels in Section 4.3 (Case Studies on NMAGs) should be included within Section 4.2 (Metal-based Aerogels (MAGs))
Comment 4
The analysis, summary, and synthesis of the literature are insufficient. For example, in Section 4.1.2 (Case Studies Concerning Some Relevant Applications of CAGs), a large number of related studies and research results are listed, but there is no comprehensive summary of the current research status and trends. Furthermore, the key issues in future research and the directions for future development have not been addressed.
Comment 5
In Sections 4.2.1 (Sol-Gel Methods to Prepare Noble Metal Aerogels (NMAGs)) and 4.2.2 (Non-Sol-Gel Methods to Prepare Metallic Aerogels), the introduction of different synthesis methods is relatively superficial, merely listing the experimental results of various methods. Relevant images or tables should be added to compare the performance of NMAGs prepared by the two different methods.
Comment 6
In Section 4.3 (Case Studies on NMAGs), it is mentioned that "The sol-gel approach manifests overwhelming advantages for adaptable synthesis of nanostructured NMAGs under mild conditions." However, no relevant references are cited to support this statement.
Comment 7
In Section 4.3 (Case Studies on NMAGs), only the synthesis of different types of NMAGs by various researchers is listed, and Table 13 only compares the synthesis methods, morphology, and composition of different types of NMAGs. However, their performance and related applications are not introduced.
Comment 8
In Section 5 (Interdisciplinarity of Aerogels), the applications of aerogels in other fields are merely listed briefly, without summarizing the current research status in this area. The main progress, controversies, and issues in existing research are not synthesized. This review fails to inspire subsequent academic research and lacks a guiding significance for the field.
Comment 9
Although this review cites a large number of references, it does not sufficiently summarize the recent applications and significant progress of aerogels in related fields in recent years.
Author Response
I believe this work will be highly valuable for researchers in the field. However, several aspects of the manuscript could be improved, I suggest its publication after major revisions.
Comment 1
In the Abstract of this paper, it is mentioned that “Specifically, the properties, specific synthesis including the one and two-step sol-gel methods, dealloying, combustion and bio-template processes, and their possible uses were disserted.” However, in the main text, the properties and specific synthesis methods are described in a very cursory manner.
I thank the Reviewer for her/his valuable comment, which has enabled me to extensively expand the part concerning the synthetic methods currently available to prepare AGs, different from chalcogenide (CAGs) and metal ones (MAGs) (lines 165-190, 212-234, 291-295, and 296-304) and those needed to specifically prepare these other types of AGs (topics of this paper) (lines 441-522 and 758-762). The microstructure characteristics and morphological properties of the as prepared AGs were extensively discussed (lines 165-190, 305-373, 522-532, 814-833 and 899-938). To this end, several new Figures (Figure S1-S17) were included in the new appositely created Supplementary Materials file. To expand this part, new Figures were included in the main text (Figure 3, 4, 5, 6 and 7). The sentence signalled by the Reviewer in the abstract was reformulated as the most part of the abstract, to be more consistent with the contents of the paper and to better explain the rational and reasons of the sections sequence followed.
Comment 2
Throughout the paper, there are numerous tables but only two figures, which makes the article relatively difficult to comprehend and may hinder readers' understanding of the content. It is recommended to add relevant images and schematic diagrams. For example, in Section 1 (Introduction), schematic diagrams of different aerogels can be included; in Section 3 (Synthetic Methods to Achieve AGs: A Brief Overview), real images of aerogels synthesized via the sol-gel method can be added.
These suggestions from the Reviewer have been very helpful in improving the quality of this study and readability of my paper. As asked, a Schematic diagram of different AGs has been included in the Introduction as Figure 1. Real images of the microstructure of AGs prepared according to all procedures described in Section 3 and 4, as well as Schemes of some of such procedures have been included in the Supplementary Materials file appositely created as described in the previous point (Figure S1-S17). Moreover, two new Schemes concerning the EA methods have been included in the main text as Figure 3 and 4, and a schematic summary of the synthetic traditional methods currently existing to prepare AG different from CAGs and MAGs has been inserted in the main text as Figure 5, as reported in the previous point. Following this suggestion, schematic summaries of the synthetic methods needed to prepare CAGs (MCAGs and MMCAGs), NMAGs and MAGs has been inserted in the main text as Figure 6 and 7, as previously mentioned.
Comment 3
The logic of this paper is unclear. In Section 3 (Synthetic Methods to Achieve AGs: A Brief Overview), the traditional sol-gel method for preparing aerogels is introduced. However, in Sections 4.2.1 (Sol-Gel Methods to Prepare Noble Metal Aerogels (NMAGs)) and 4.2.2 (Non-Sol-Gel Methods to Prepare Metallic Aerogels), the sol-gel and non-sol-gel methods for preparing metal-based aerogels are discussed again. Similar content should be integrated into Section 3. Additionally, the introduction of noble metal aerogels in Section 4.3 (Case Studies on NMAGs) should be included within Section 4.2 (Metal-based Aerogels (MAGs))
The suggestion of Reviewer to move Section 4.2.1 and 4.2.2 in Section 3 is understandable and could appear rational. Anyway, synthetic methods described in Section 3 are different from those described in Sections 4.2.1. and 4.2.2., which are instead specific processes needed to prepare noble metal/metal AGs and therefore discussed in Section 4.2., which focus on them. Similarly, methods needed to prepare CAGs (MCAGs and MMCAGs), whose part has been expanded in the revised manuscript, have been included in Section 4.1 focused on them. Traditional and improved sol-gel methods (EA and DIS) and nanoparticles-based methods to prepare other AGs, described in Section 3, are not suitable to achieve CAGs as well as metal AGs (MAGs), but need the substantial modifications described in Sections 4.1.1., Section 4.2.1 and Section 4.2.2. As above-mentioned, I have rationally not reported methods described in paper in these Sections previously, with the others in Section 3, since they are specific for CAGs and MAGs, which have Sections 4.1 and 4.2., as dedicated ones. As asked, Section 4.3 (Case Studies on NMAGs) has been included within Section 4.2 (Metal-based Aerogels (MAGs), as Section 4.2.3. Please, see line 939.
Comment 4
The analysis, summary, and synthesis of the literature are insufficient. For example, in Section 4.1.2 (Case Studies Concerning Some Relevant Applications of CAGs), a large number of related studies and research results are listed, but there is no comprehensive summary of the current research status and trends. Furthermore, the key issues in future research and the directions for future development have not been addressed.
I thank the Reviewer for having signalled this lack of information. A summary of the current research status and trends and of the key issues in future research and the directions for future development of CAGs, intended as MCAGs and MMCAGs has been included in the new Section 4.1.3. “Authors Summary and Considerations on CAGs”. Please, see lines 664-727.
Comment 5
In Sections 4.2.1 (Sol-Gel Methods to Prepare Noble Metal Aerogels (NMAGs)) and 4.2.2 (Non-Sol-Gel Methods to Prepare Metallic Aerogels), the introduction of different synthesis methods is relatively superficial, merely listing the experimental results of various methods. Relevant images or tables should be added to compare the performance of NMAGs prepared by the two different methods.
As already reported in previous points, Section 4.2.1. and 4.2.2. have been extensively modified and expanded adding schematic summaries of synthetic methods needed to prepare NMAGs and MAGs, as new Figure 6 and 7 in the main text. Relevant images of morphology and microstructure of NMAGs and MAGs prepared by the described methods have been included in Supplementary Materials (Figure S10-S17) and have been discussed in the main text. Please, see lines 441-522, 522-532, 758-762, 814-833, 899-938.
Comment 6
In Section 4.3 (Case Studies on NMAGs), it is mentioned that "The sol-gel approach manifests overwhelming advantages for adaptable synthesis of nanostructured NMAGs under mild conditions." However, no relevant references are cited to support this statement.
The reference supporting the statement has been included. Ref. 272 (line 943).
Comment 7
In Section 4.3 (Case Studies on NMAGs), only the synthesis of different types of NMAGs by various researchers is listed, and Table 13 only compares the synthesis methods, morphology, and composition of different types of NMAGs. However, their performance and related applications are not introduced.
I thank the Reviewer for having evidenced that applications and performances for NMAGs and MAGs reported in the original paper were limited thus seeming absent. So, despite some applications of the most notable NMAGs and MAGs were already reported in the text of the not revised paper (parts highlighted in yellow), additional ones were inserted. Please, see red paragraphs and red paragraphs evidenced in yellow in lines 952-953, 959-960, 964-966, 970-974, 978-980, 982-985, 986-991, 994-995, 1001, 1040-1041, 1044-1048, 1060-1065, 1082-1085 and 1103-1112.
Comment 8
In Section 5 (Interdisciplinarity of Aerogels), the applications of aerogels in other fields are merely listed briefly, without summarizing the current research status in this area. The main progress, controversies, and issues in existing research are not synthesized. This review fails to inspire subsequent academic research and lacks a guiding significance for the field.
On the valuable suggestion by the Reviewer, Section 5 has been expanded adding subsection 5.1., titled “Current Research Hotspots and Future Development Trends” which should meet the Reviewer’s request. Please, see lines 1165-1195.
Comment 9
Although this review cites a large number of references, it does not sufficiently summarize the recent applications and significant progress of aerogels in related fields in recent years.
I hope that, following all suggestions from the Reviewer and applying them also to parts not directly indicated by her/him, but considered by me needed of improvement and/or expansion, and the extensive work of revision carried out, the resulting manuscript could meet the complacency of the Reviewer and that also the criticisms signalled in this point could be addressed. I thank again the Reviewer for her/his help.
The English could be improved to more clearly express the research.
On suggestion of the Reviewer, the manuscript has been revised by prof. Deirdre Kantz, English mother tongue teacher, my colleague, who works for University of Genoa and Pavia, to improve language.
Reviewer 2 Report
Comments and Suggestions for Authors
The manuscript is the second part of the review paper series on aerogels. The current part outlines the research on chalcogenides and metal-based aerogels. I cannot evaluate the first part because it’s still unpublished; comments will focus on the current paper. In my opinion, the critical aspect in the current review is too weak. Unfortunately, the recent aerogel research community, and more or less all other scientific communities, has been suffering from an influx of unreliable research including inappropriate overexaggeration and fraudulent novelty claims, eg, poor reliability of thermal conductivity data (J. Sol-Gel Sci. Technol. 109 (2024) 569, Angew. Chem. Int. Ed. 64 (2025) e202504250). To maintain the integrity and credibility of the academic community, review papers must be critical to some extent. The manuscript should be rechecked and revised with quantitative evaluation of the picked-up papers, not blindly accepting "excellent" "high-performance" of the original papers. For details:
1. Application potential should be critically discussed with comparison to non-aerogel materials.
Section 4.1.2: To the best of my knowledge, chalcogenides-based aerogels are not major contenders for ion/dye adsorbents. Please evaluate their potential compared to other aerogels and non-aerogel commercial porous materials/polymer-based materials.
Section 4.3: Metal aerogels have been extensively investigated as potential catalysts for these ~30 years, and their industrialization is still in the early stage. Potential for electrocatalysts of metal-based aerogels should be compared to other metal-supporting practical catalysts.
2. As the title says, chalcogenides- and metal-based aerogels have not been in the mainstream of aerogel research compared to SiO2 and related oxides/carbides. The manuscript should discuss the disadvantage of chalcogenides- and metal-based aerogels, eg, cost, complex synthesis methods, stability, which has been inhibiting their industrialization.
3. The title says "less patented". Please provide a brief introduction to the intellectual property situation of these aerogels, or the title should be reconsidered.
4. Even though there is no common definition of aerogels, drying methods-microstructure feature of aerogels should be mentioned. The majority of the current manuscript focuses on mesoporous aerogels, mainly prepared by supercritical or ambient drying. The following fact should be mentioned: we have a flood of freeze-dried "aerogels" in recent literature, which have >micrometer pores and totally different nature/function from those in the majority of the current manuscript.
5. Non-noble metal aerogels should be briefly explained, eg, the early stage of metal aerogels research had focused on Fe.
6. Please shorten Section 4.3 to make it more concise.
7. Table 14: The industrial applications should be classified with their stages, eg, widely commercialized/industrialization in progress/only emerging in academic papers.
Author Response
The manuscript is the second part of the review paper series on aerogels. The current part outlines the research on chalcogenides and metal-based aerogels. I cannot evaluate the first part because it’s still unpublished; comments will focus on the current paper.
The Reviewer is right. Aerogel Part 1 was not published during the first-round revision for Aerogel Part 2, for which the Reviewer was asked to work for revision. Anyway, the Preprints of paper “Part 1” were already available online and consultable at Alfei, S. Aerogels Part 1. A Focus on the Most Patented Ultralight, Highly Porous Inorganic Networks and the Plethora of Their Advanced Applications. Preprints 2025, 2025080009. https://doi.org/10.20944/preprints202508.0009.v1, as reported in the reference list (Ref. 50). However, I assure the Reviewer, that despite the same general area of interest, i.e. AGs, the two reviews are independent of each other and can be considered individually, without knowing the details of the other. In this regard, I thank the Reviewer for basing her/his comments solely on this Part 2 review, on which she/he was asked to judge, while it is my pleasure let her/him know that Aerogel Part 1 has been accepted for publication just today (03 September) and will be cited in this one (if accepted) as published article.
In my opinion, the critical aspect in the current review is too weak. Unfortunately, the recent aerogel research community, and more or less all other scientific communities, has been suffering from an influx of unreliable research including inappropriate overexaggeration and fraudulent novelty claims, eg, poor reliability of thermal conductivity data (J. Sol-Gel Sci. Technol. 109 (2024) 569, Angew. Chem. Int. Ed. 64 (2025) e202504250). To maintain the integrity and credibility of the academic community, review papers must be critical to some extent.
I thank the Reviewer for these comments, which made me think a lot about the reliability of information reported in this paper and its robustness. In this regard, I have considered both the interesting and relevant articles suggested by the Reviewer, establishing the unreliability and fraudulence of several in formation findable in literature concerning AGs. Anyway, I found that (as said by the Reviewer her/himself), both articles contest and find unrealistic and fraudulent several reported data concerning thermal conductivity. For this, the assumption of the Reviewer cannot be applied to data reported here, which I consider not suffering from an influx of unreliable research, including inappropriate overexaggeration and fraudulent novelty claims. In fact, while the reported fraudulent data regard thermal conductivity, which is a property peculiar of silica AGs (SAGs) and metal oxide AGs (MOAGs), minimally considered in my paper, the present paper reports information and data on chalcogenide (CAGs) and metal (MAGs) AGs. Not concerning data on thermal conductivity of CAGs and MAGs, they regard their efficiency (at least in laboratory experimentation), in environmental remediation (removal of heavy metal ions, radionucleotides, dyes), energy production, gas adsorption, gas separation, catalysis, electrocatalysis, photoelectric catalysis, electrochemical catalysis, as well as their potential application as conductors and semiconductors, capacitors, sensors, fuel cells and batteries. Only, in Section 1 and 5, I mentioned the high performance of SAG in thermal insulation due to their low thermal conductivity, but reporting real applications in aerospace industry (NASA), which are unlikely to be unreal or exaggerated. Moreover, the second article suggested by the Reviewer, minimally cites the topics of my review, and no unreliable data for these materials has been evidenced and the most correct attribution of the name AGs to materials has been discussed.
The manuscript should be rechecked and revised with quantitative evaluation of the picked-up papers, not blindly accepting "excellent" "high-performance" of the original papers.
This has been done.
For details:
- Application potential should be critically discussed with comparison to non-aerogel materials.
Section 4.1.2: To the best of my knowledge, chalcogenides-based aerogels are not major contenders for ion/dye adsorbents. Please evaluate their potential compared to other aerogels and non-aerogel commercial porous materials/polymer-based materials.
Section 4.3: Metal aerogels have been extensively investigated as potential catalysts for these ~30 years, and their industrialization is still in the early stage. Potential for electrocatalysts of metal-based aerogels should be compared to other metal-supporting practical catalysts.
I thank the Reviewer for having evidenced that applications and performances for CAGs, NMAGs and MAGs reported in the original paper were limited thus seeming absent, as well as the comparisons of their efficiency with that of other AG and not AG materials. So, despite some applications of the most notable CAGs, NMAGs and MAGs and some comparisons were already reported in the text (parts highlighted in yellow) of the not revised paper, additional ones were inserted. Rationally, considering that this paper should focus on chalcogenide AGs (MCAGs and MMCAGs) and metal AGs (NMAGs and MAGs), we reported and critically discussed mainly the potential applications of these materials and more sporadically those of other AGs. Application potential of MCAGs and MMCAGs have been reported in red and highlighted in yellow or only in red in lines 142-145, 399-409, 415-423, 425-426, Table 3 (last row), lines 458-461, 464-467, 469-472 and475-476, in all Section 4.1.2. (lines 533-663) containing Tables 4-12, also in part not in red or evidenced in yellow. Specifically, Table 4-12 report the potential of MCAGs in environmental remediation (heavy metals and dyes removal by adsorption, as well as radionucleotide management by ion exchange mechanisms). The efficiency of MCAGs in iodine adsorption and radionucleotides remediation has been disserted also in lines 618-663. Several comparisons between the efficiency of MCAGs and MMCAGs in environmental remediation (removal of heavy metals and dyes) with that of other materials had been already reported in Tables 5 and 9, in the original version of the work (evidenced in yellow). Anyway, to satisfy the Reviewer, additional comparisons between the efficiency of MCAGs in removing iodine and radionucleotides and that of other types of AGs have been added in lines 645-663. Additionally, a critical discussion on current research status and trends of CAGs with key issues in future research and directions for their future development has been inserted as Section 4.1.3 (lines 664-727). Concerning potential applications of MAGs in catalysis, electrocatalysis, energy materials and for the manufacture of cathode for batteries and some comparisons have been reported in lines 952-953, 959-960, 964-966, 970-974, 978-980, 982-985, 986-991, 994-995, 1001, 1040-1041, 1044-1048, 1060-1065, 1082-1085 and 1103-1112.
- As the title says, chalcogenides- and metal-based aerogels have not been in the mainstream of aerogel research compared to SiO2 and related oxides/carbides. The manuscript should discuss the disadvantage of chalcogenides- and metal-based aerogels, eg, cost, complex synthesis methods, stability, which has been inhibiting their industrialization.
The information required by the Reviewer has been reported and discussed in lines 54-82, 381-389 and in new Section 4.3.1.
- The title says "less patented". Please provide a brief introduction to the intellectual property situation of these aerogels, or the title should be reconsidered.
The nature of this Title is linked to that of Part 1 paper consultable in Ref. 50., Aerogels Part 1. A Focus on the Most Patented Ultralight, Highly Porous Inorganic Networks and the Plethora of Their Advanced Applications. Preprints 2025, 2025080009. https://doi.org/10.20944/preprints202508.0009.v1. The choice of these titles is justified by data reported in the CAS Content Collection (Figure 2 in our recent paper [50]), where while silica AGs and metal oxide AGs appear as the main experimented to produce patents and are on the market form years, CAGs and MAGs are still little studied and mainly journal articles have been published. Patents are rare and this evidences that their translation on the marketed still is faraway. This explanation is reported at the beginning of Section 4 (lines 375-389).
- Even though there is no common definition of aerogels, drying methods-microstructure feature of aerogels should be mentioned. The majority of the current manuscript focuses on mesoporous aerogels, mainly prepared by supercritical or ambient drying. The following fact should be mentioned: we have a flood of freeze-dried "aerogels" in recent literature, which have >micrometer pores and totally different nature/function from those in the majority of the current manuscript.
Drying methods-microstructure feature of aerogels have been mentioned and discussed in the main text in lines 165-191 and related images showing the different microstructure of AGs depending on drying methods used have been included in Supplementary Materials as Figure S1.
- Non-noble metal aerogels should be briefly explained, eg, the early stage of metal aerogels research had focused on Fe.
Several examples of non-noble metal AGs had been reported in Section 4.2, already in the not-revised paper. Please, consider carefully Sections 4.2.1, 4.2.2. and 4.2.3 (numbering changed in the revised manuscript). Concerning iron-based AGs, the Reviewer can find information in lines 151-153, 736-748 and 1082-1085.
- Please shorten Section 4.3 to make it more concise.
I kindly ask the Reviewer to not force me to shorten this Section (now Section 4.2.3) to not oppose the requests from the other Reviewer who asked for additional details.
- Table 14: The industrial applications should be classified with their stages, eg, widely commercialized/industrialization in progress/only emerging in academic papers.
The required information has been provided in the footnotes of the Table.
The English is fine and does not require any improvement.
We thank the Reviewer for this positive comment on my English.
Round 2
Reviewer 1 Report
Comments and Suggestions for Authors
I have reviewed the revised manuscript submitted by the author in response to the first round of review comments, as well as the detailed Point-to-Point Response. The author has provided comprehensive, in-depth, and high-quality responses to all questions and suggestions raised by the reviewers (including myself). The quality of the revised manuscript has been significantly improved, and all key concerns have been addressed.
I am extremely satisfied with the author's revision work and believe that the manuscript now meets the publication standards. Therefore, I recommend accepting this manuscript.
Author Response
I thank a lot this Reviewer for her/his positive comments and for having appreciated both the work made by me for meeting her/his requests and comments, but also that made to address comments by other Reviewers.